# Alkyne-tagged SERS nanoprobe for understanding Cu$^+$ and Cu$^{2+}$ conversion in cuproptosis processes

Sihan Zhang[1], Yuxiao Mei[1], Jiaqi Liu[1], Zhichao Liu[1] ✉ & Yang Tian ®[1] ✉

Simultaneously quantifying mitochondrial Cu$^+$ and Cu$^{2+}$ levels is crucial for evaluating the molecular mechanisms of copper accumulation-involved pathological processes. Here, a series of molecules containing various diacetylene derivatives as Raman reporters are designed and synthesized, and the alkyne-tagged SERS probe is created for determination Cu$^+$ and Cu$^{2+}$ with high selectivity and sensitivity. The developed SERS probe generates well-separated distinguishable Raman fingerprint peaks with built-in corrections in the cellular silent region, resulting in accurate quantification of Cu$^+$ and Cu$^{2+}$. The present probe demonstrates high tempo-spatial resolution for real-time imaging and simultaneously quantifying mitochondrial Cu$^+$ and Cu$^{2+}$ with long-term stability benefiting from the probe assembly with designed Au-C≡C groups. Using this powerful tool, it is found that mitochondrial Cu$^+$ and Cu$^{2+}$ increase during ischemia are associated with breakdown of proteins containing copper as well as conversion of Cu$^+$ and Cu$^{2+}$. Meanwhile, we observe that parts of Cu$^+$ and Cu$^{2+}$ are transported out of neurons by ATPase. More importantly, cuproptosis in neurons is found including the oxidative stress process caused by the conversion of Cu$^+$ to Cu$^{2+}$, which dominates at the early stage (<9 h), and subsequent proteotoxic stress. Both oxidative and proteotoxic stresses contribute to neuronal death.

Real-time monitoring of active substances within mitochondria holds significant importance in understanding the molecular mechanisms of oxidative stress-involved physiological and pathological processes. The mitochondrion is an important copper store in cells, whose physiological functions are closely related to copper ions homeostasis[1–3]. As critical redox-active substances in the mitochondrion, copper ions have two forms: reduced state (Cu$^+$) and oxidized state (Cu$^{2+}$). Cuprous ion (Cu$^+$) could be converted to cupric ion (Cu$^{2+}$) by triggering the conversion of molecular oxygen into reactive oxygen species (ROS)[4,5], while Cu$^{2+}$ can be reduced to Cu$^+$ by intracellular endogenous antioxidants[6]. Nevertheless, the disturbance of Cu$^+$ and Cu$^{2+}$ levels would break the homeostasis of oxidative stress in mitochondria,

which may further cause massive ROS, as well as damage to DNA, proteins and lipids[7–10]. Therefore, instantaneous imaging and quantitative analysis of Cu$^+$ and Cu$^{2+}$ simultaneously in mitochondria were vitally important for understanding the molecular mechanism of mitochondria-related biological events.

In the past several decades, many methods have been widely used to detect either Cu$^{2+}$ or Cu$^+$, such as atomic absorption spectrometry (AAS)[11], mass spectrometry (MS)[12], fluorescent methods[13–15], and electrochemical methods[16]. We have designed and developed several ratiometric fluorescent and electrochemical sensors to monitor and accurately quantify either Cu$^{2+}$ or Cu$^+$ in the live brain or living cells with high selectivity[17–20]. However, simultaneous quantitative analysis

[1]State Key Laboratory of Molecular & Process Engineering, School of Chemistry and Molecular Engineering, East China Normal University, Dongchuan Road 500, Shanghai, China. ✉e-mail: zcliu@chem.ecnu.edu.cn; ytian@chem.ecnu.edu.cn

of Cu$^+$ and Cu$^{2+}$ is quite difficult. Surface-enhanced Raman spectroscopy (SERS) offers highly sensitive molecular fingerprint information, as well as resistance to autofluorescence and photobleaching[21,22]. Recently, an SERS probe was developed by our group for simultaneous sensing of extracellular Cu$^+$ and Cu$^{2+}$ in mouse cerebral cortex, and three routes were proposed for explaining the mechanisms of Cu$^+$ and Cu$^{2+}$ increases during ischemia[23]. However, whether the increase in extracellular copper ions originated from inside the live cells remains unclear. More importantly, the contributions of Cu$^+$ and Cu$^{2+}$ to cell death in processes related to Cu accumulation, such as ischemia and cuproptosis[24–27], has not yet been elucidated, due to the lack of powerful tools for sensing and quantitative analysis of Cu$^+$ and Cu$^{2+}$ simultaneously, especially in mitochondria. To draw a clear picture of the mechanisms of copper accumulation-related processes, the development of an effective probe for simultaneous analysis of mitochondrial Cu$^+$ and Cu$^{2+}$ was needed, particularly in mitochondria. However, SERS probes assembled based on Au-S bonds are easily susceptible to unpredictable aggregation in cells[28–30]. In addition, the SERS response signal below 1100 cm$^{-1}$ could be easily affected by native species in the complex cellular environment[31,32]. It is therefore desirable to develop a single SERS probe for real-time imaging and accuracy qualifying of intracellular Cu$^+$ and Cu$^{2+}$ simultaneously, especially in mitochondria.

In this work, a single alkyne-tagged SERS probe is designed and developed for simultaneously instantaneous sensing and imaging of Cu$^+$ and Cu$^{2+}$ in mitochondria. Firstly, a series of molecules (named as Cu$^1$R$_X$ and Cu$^2$R$_X$) containing different diacetylene derivatives as Raman reporters at cellular Raman-silent region are designed. By using theoretical calculations, 4-((4-((bis (2-((2-(ethylthio) ethyl) thio) ethyl) amino) methyl) phenyl) buta-1, 3-diyn-1-yl)-N-(prop-2-yn-1-yl) benzamide (Cu$^1$R$_5$) and N, N-bis (pyridin-2-ylmethyl) octa-2, 4, 7-triyn-1-amine (Cu$^2$R$_1$) are determined as the optimized Raman molecules for specific recognition of Cu$^+$ and Cu$^{2+}$ with high sensitivity and clearly distinguishable Raman signals in cellular silent region, respectively. Meanwhile, 1-ethynyl-4-[2-[tris(1-methylethyl) silyl] ethynyl] benzene (EETP) exhibiting distinct Raman signals within cellular silent regions is used as reference elements, which improves the accuracy for quantifying Cu$^+$ and Cu$^{2+}$, and [4-oxo-4-(2-propyn-1-ylamino) butyl] triphenylphosphonium (TPP) is employed for targeting mitochondria. In addition, gold nanostars (GNs) are synthesized as SERS substrates to improve the sensitivity. Then, Cu$^1$R$_5$, Cu$^2$R$_1$, EETP and TPP molecules are assembled on GNs through Au−C≡C bond to prepare the SERS probe, denoted as CuPM@GN. The developed CuPM@GN probe exhibits distinguishable Raman responses toward Cu$^+$ and Cu$^{2+}$ at 2213 cm$^{-1}$ and 2238 cm$^{-1}$, respectively, with a reference Raman signal at 2155 cm$^{-1}$ for built-in calibration. Besides, the developed probe shows remarkable selectivity and good accuracy with high temporal resolution of less than ~16 s. Benefiting from the developed probe with good reproducibility and stability, as well as excellent biocompatibility, mitochondrial Cu$^+$ and Cu$^{2+}$ are quantified simultaneously, and it is discovered that the levels of mitochondrial free Cu$^+$ and Cu$^{2+}$ in neurons increase significantly during ischemia, especially Cu$^{2+}$. More importantly, we find that the destruction of the structure of copper-containing proteins as well as Cu$^+$ and Cu$^{2+}$ conversion are two key pathways for the increase of mitochondrial copper ions during ischemia, while the elevated free copper ions in neurons are partially transported out of neurons by ATPase. Furthermore, we find that cuproptosis also occurs in neurons during copper overload. More importantly, it is discovered that oxidative stress (O$_2$$^{·-}$ burst) caused by the conversion of Cu$^+$ to Cu$^{2+}$ in mitochondria is the main event at the early stage (<9 h) of cuproptosis, thereby partially exacerbating proteotoxic stress-dominated event at the later stage of cuproptosis. Both oxidative stress and proteotoxic stress contribute to neuronal death during cuproptosis.

## Results

### Design and synthesis of a single alkyne-tagged SERS probe for detection of Cu$^+$ and Cu$^{2+}$ simultaneously in mitochondria with high selectivity

As a starting point of this work, different Raman molecules for specific identification of Cu$^+$ or Cu$^{2+}$ were designed based on chemical reactions and characteristic fingerprint peaks involving dual-recognition strategies, named Cu$^1$R$_X$ and Cu$^2$R$_X$ (Fig. 1a). All these molecules include three parts: the recognition groups for specifically capturing Cu$^+$ or Cu$^{2+}$, diacetylene derivatives (R$_1$–R$_5$) as Raman reporters at cellular Raman-silent region, and terminal alkynyl groups for stable assembly through Au−C≡C bond. Raman spectra of these molecules were first obtained by theoretical calculations, and Raman bands assignment of these molecules before and after addition of Cu$^+$ or Cu$^{2+}$ were obtained (Supplementary Tables 1–10), especially the characteristic peaks attributed to stretching vibrations (C≡C) of C≡C−C≡C group. As shown in Fig. 1b, changing the alkyls at the end of diacetylene derivatives led to an obvious red shift of the Raman shift of these molecules, and the Raman shift belongs to stretching vibrations (C≡C) of C≡C−C≡C group in these molecules redshifted by -23 cm$^{-1}$ after the alkyls of diacetylene derivative (R$_1$) were replaced by benzyl and benzamide simultaneously (R$_5$). These results may be due to the benzene ring weakens the stretching force constant of the vibrational mode by influencing π-electron delocalization on the diacetylene[23], promising that the Raman vibrational frequencies of diacetylene group depend on the chemical structures. To achieve simultaneous detection of Cu$^+$ and Cu$^{2+}$, Cu$^1$R$_5$ and Cu$^2$R$_1$ were selected for specific recognition of Cu$^+$ and Cu$^{2+}$, with calculated Raman peaks at -2213 cm$^{-1}$ and -2238 cm$^{-1}$, respectively. The structures of Cu$^1$R$_5$ and Cu$^2$R$_1$ were synthesized and characterized by high-resolution mass spectrum (HR-MS), $^1$H nuclear magnetic resonance (NMR), and $^{13}$C NMR (Supplementary Figs. 1–16).

Then, for simultaneous detection of Cu$^+$ and Cu$^{2+}$ in mitochondria, an individual alkyne-tagged SERS probe was designed and synthesized by conjugation of Cu$^1$R$_5$, Cu$^2$R$_1$, [4-oxo-4-(2-propyn-1-ylamino) butyl] triphenylphosphonium (TPP), and 1-ethynyl-4-[2-[tris(1-methylethyl) silyl] ethynyl] benzene (EETP) onto gold nanostars (GNs) through Au-C≡C bond with high stability, which was denoted as CuPM@GN (Fig. 1c). Among the CuPM@GN, EETP was used as a built-in correction for accurate quantitative analysis, while TPP was set as a mitochondria-targeted molecule. In addition, GNs were selected as SERS substrate for enhancing signals. Transmission electron microscopy (TEM) results demonstrated that the synthesized GNs were anisotropic nanoparticles with a mean diameter of 48 ± 6 nm ($n = 20$ nanoparticles, S.D.) and abundant sharp branches, (Supplementary Fig. 17a). No significant increase in size after CuPM@GN was formed (Supplementary Fig. 17c). In addition, the maximum absorption wavelength of GNs showed in the UV-vis absorption spectrum was -700 nm (Supplementary Fig. 17b), which was slightly red-shifted (-3 nm) after CuPM@GN was formed (Supplementary Fig. 17d). These results suggested that CuPM@GN was dispersed without significant agglomeration, which was also confirmed by zeta potential and dynamic light scattering (DLS) results (Supplementary Fig. 18).

In addition, the calculated Raman enhancement factor (EF) value for GNs was approximately $7.42 \times 10^6$ (Supplementary Fig. 19). Interestingly, after Cu$^1$R$_5$, Cu$^2$R$_1$, EETP and TPP were assembled onto GNs to form CuPM@GN, four typical Raman peaks were observed located at -2213 cm$^{-1}$, -2238 cm$^{-1}$, -2155 cm$^{-1}$ and -1027 cm$^{-1}$, which were attributed to stretching vibrations of conjugated C≡C bonds in Cu$^1$R$_5$, stretching vibrations of conjugated C≡C bonds in Cu$^2$R$_1$, stretching vibrations of C≡C bond in EETP and stretching vibrations of benzene ring in TPP, respectively, demonstrating the successful assembly of CuPM@GN probe. Moreover, there was no cross-talk observed for the characteristic Raman peaks of these molecules (Fig. 1d), indicating the

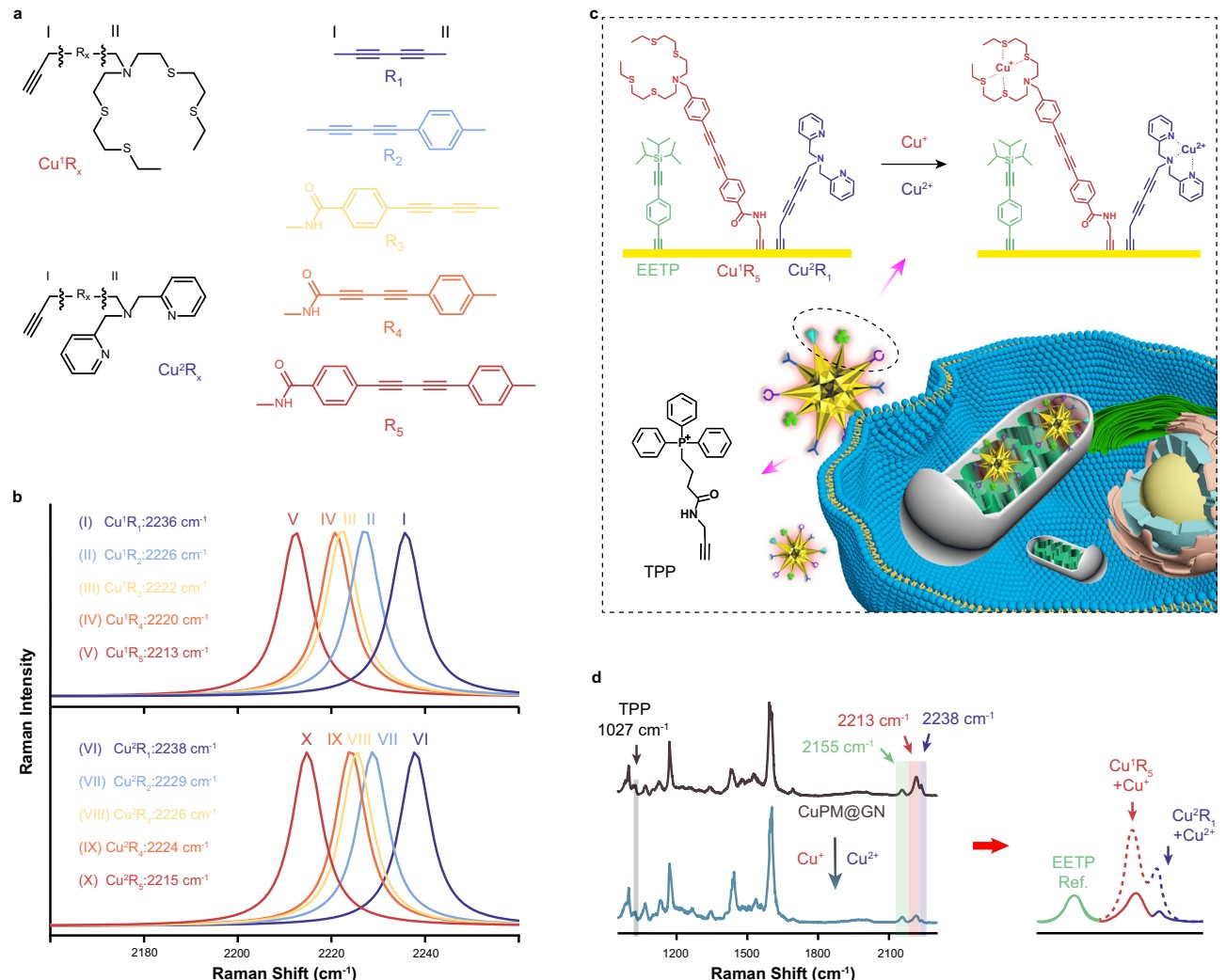

**Fig. 1 | Design of SERS probe for simultaneous recognition of Cu⁺ and Cu²⁺.**
**a** Design principles and structures of the diacetylene-labeled Raman molecules ($Cu^1R_x$ and $Cu^2R_x$) for recognition of $Cu^+$ and $Cu^{2+}$, respectively. $R_1$ to $R_5$ indicated different diacetylene derivatives (right). **b** Raman spectra of $Cu^1R_x$ (top) and $Cu^2R_x$ (bottom) were obtained by theoretical simulation, respectively. **c** Schematic diagram of the designed CuPM@GN probe for simultaneous determination of $Cu^+$ and $Cu^{2+}$ in mitochondria. **d** SERS spectra of the developed CuPM@GN probe before and after the addition of $Cu^+$ and $Cu^{2+}$ simultaneously.

developed SERS probe could determine $Cu^+$ and $Cu^{2+}$ simultaneously based on the separated Raman signals (Supplementary Table 11).

**SERS determination for CuPM@GN toward $Cu^+$ and $Cu^{2+}$ in vitro**
To further assess the detection capability of the developed CuPM@GN probe, we initially conducted in vitro experiments with simultaneous titration of $Cu^+$ and $Cu^{2+}$ in mitochondrial lysates. As shown in Fig. 2a, the peak intensity at 2213 cm⁻¹ ($I_{2213}$) decreased with the gradual addition of $Cu^+$, while that at 2155 cm⁻¹ ($I_{2155}$) hardly changed. The SERS intensity ratio between $I_{2213}$ and $I_{2155}$ ($I_{2213}/I_{2155}$) showed good linearity with the concentrations of $Cu^+$ in the range of 0.50–14.00 μM. The detection limit (LOD) was estimated to be 0.32 ± 0.03 μM (3σ/S, where S is the slope of calibration curve, and σ is the standard deviation (S.D.) of the probe sample, $n = 5$) (Fig. 2d). On the other hand, as demonstrated in Fig. 2b, the peak intensity at 2238 cm⁻¹ ($I_{2238}$) decreased with increasing concentration of $Cu^{2+}$, yet $I_{2155}$ still remained unchanged. $I_{2238}/I_{2155}$ exhibited a good linear relationship with the concentrations of $Cu^{2+}$ from 0.50 to 16.00 μM, with a LOD of 0.44 ± 0.02 μM (3σ/k) ($n = 5$, S. D., Fig. 2e). In addition, simultaneous determination of $Cu^+$ and $Cu^{2+}$ in vitro was also achieved by using CuPM@GN. With increasing concentrations of $Cu^+$ and $Cu^{2+}$, both $I_{2213}$ and $I_{2238}$ went down, while $I_{2155}$ hardly changed (Fig. 2c). The slope coefficients of the

calibration curves in Fig. 2f, which were used for the simultaneous determination of $Cu^+$ and $Cu^{2+}$, agrees well with those depicted in Fig. 2d, e. This consistency suggests that there was no cross-talk occurred during the determination of $Cu^+$ and $Cu^{2+}$ simultaneously. Moreover, hydrodynamic diameters of individual CuPM@GN probe were only increased by 21.84%–49.43% with increasing concentrations of copper ions (Supplementary Fig. 20), which were smaller than those reported in the literatures (>-115%)[33–35], proving no obvious aggregation was observed for CuPM@GN probe after addition of copper ions. These results demonstrated good stability of our developed CuPM@GN probe during copper ions sensing.

Then, the recognition mechanisms of the developed Raman probe toward $Cu^+$ and $Cu^{2+}$ and the Raman spectra of the products resulting from the chemical reactions were evaluated. Stoichiometry compositions of $Cu^1R_5$-$Cu^+$ complex and $Cu^2R_1$-$Cu^{2+}$ complex were first determined by absorbance spectra measurements. A new band around 386 nm was observed after addition of $Cu^+$ (Supplementary Fig. 21a), it can be attributed to the extension of the electron conjugation system by the chelate structure formed after metal ion coordinated with the ligand, resulting in the emergence of new absorption peaks at longer wavelengths[36]. In addition, an iso-absorption point (-330 nm) was observed between the peak maximum of $Cu^1R_5$ (312 nm) and the peak

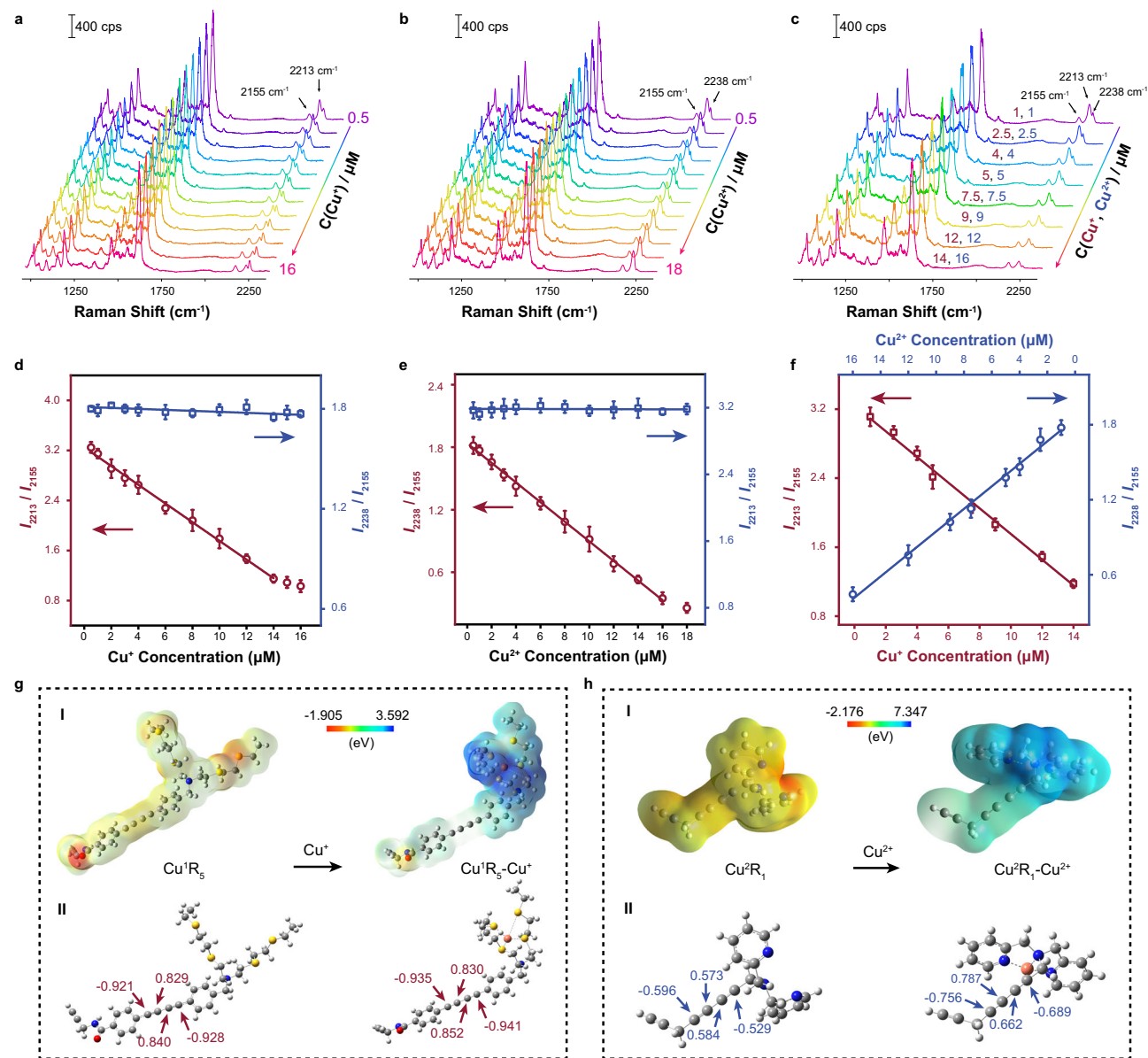

**Fig. 2 | Sensing performance of the developed SERS probes. a** SERS spectra of CuPM@GN in mitochondrial lysates with the addition of Cu⁺ with various concentrations (0.5, 1, 2, 3, 4, 6, 8, 10, 12, 14, 15, 16 μM). **b** SERS spectra of CuPM@GN in mitochondrial lysates with the addition of Cu²⁺ with various concentrations (0.5, 1, 2, 3, 4, 6, 8, 10, 12, 14, 16, 18 μM). **c** SERS spectra of CuPM@GN in mitochondrial lysates with the addition of both Cu⁺ (1, 2.5, 4, 5, 7.5, 9, 12, 14 μM) and Cu²⁺ (1, 2.5, 4, 5, 7.5, 9, 12, 16 μM) with various concentrations. **d** Plots of SERS intensity ratios ($I_{2213}/I_{2155}$ and $I_{2238}/I_{2155}$) versus the concentration of Cu⁺. Data are presented as mean ± S.D. Error bars: S.D., $n = 5$ independent experiments. **e** Plots of SERS intensity ratios ($I_{2238}/I_{2155}$ and $I_{2213}/I_{2155}$) versus concentration of Cu²⁺. Data are presented as mean ± S.D. Error bars: S.D., $n = 5$ independent experiments. **f** Calibration plots of $I_{2213}/I_{2155}$ and $I_{2238}/I_{2155}$ versus concentrations of Cu⁺ and Cu²⁺. Data are presented as mean ± S.D. Error bars: S.D., $n = 5$ independent experiments. **g** ESP (I) and Mulliken atomic charges (II) of Cu¹R₅ (left) and complex Cu¹R₅-Cu⁺ (right). **h** ESP (I) and Mulliken atomic charges (II) of Cu²R₁ (left) and complex Cu²R₁-Cu²⁺ (right). Source data are provided as a Source Data file.

maximum of the complex (386 nm), suggesting a single equilibrium step[37]. More importantly, when Cu⁺ concentration increased from 0 to 20.0 μM, the intensity of the absorption band around 386 nm was gradually increased. No obvious spectrophotometric change was observed after the concentration of Cu⁺ was higher than 20.0 μM. The stoichiometry ratio between Cu⁺ and Cu¹R₅ ligand was determined as 1:1 (Supplementary Fig. 21b), consistent with the result obtained from matrix-assisted laser desorption ionization (MALDI) (m/z peaks of 671.130) (Supplementary Fig. 22a). The binding constant of Cu¹R₅-Cu⁺ complex was determined as $(5.65 \pm 0.57) \times 10^4$ M⁻¹ (Supplementary Fig. 21c). Similarly, the stoichiometry ratio between Cu²⁺ ion and Cu²R₁ ligand was also determined as 1:1 (Supplementary Figs. 21d, e), consistent with the result obtained from MALDI (m/z peaks of 362.073)

(Supplementary Fig. 22b). The binding constant of Cu²R₁ with Cu²⁺ was calculated as $(5.61 \pm 0.94) \times 10^4$ M⁻¹ (Supplementary Fig. 21f).

Then, Mulliken atomic charges of Cu¹R₅ and Cu²R₁ were estimated before and after coordinating with the corresponding ions, for evaluating polarizability changes in the system, because polarizability is a critical factor influencing Raman intensity[38]. As shown in Fig. 2g, after Cu¹R₅ was coordinated with Cu⁺, the charges on the C atom of diacetylene group nearest to Cu⁺ changed from −0.928 to −0.941, while the charges on the C atom of diacetylene group furthest to Cu⁺ changed from −0.921 to −0.935. The difference in Mulliken charge distribution in Cu¹R₅ and Cu¹R₅-Cu⁺ complex may be attributed to the Cu⁺-induced electron deflection on diacetylene groups. Notably, the average of the difference between the number of charges on the C atom at both ends

of the C ≡ C bonds in diacetylene group changed from 1.779 to 1.793, which indicated that the strong electron acceptor Cu$^+$ induced a significant increase in the polarity of diacetylene groups[39]. Therefore, the Raman intensity at 2213 cm$^{-1}$ of Cu$^1$R$_5$ decreased in the presence of Cu$^+$. Similarly, as for Cu$^2$R$_1$ molecule, we found that the charge on the C atom of diacetylene group adjacent to N changed from −0.529 to −0.689 and that on the C atom furthest to Cu$^{2+}$ changed from −0.596 to −0.756, after Cu$^{2+}$ was coordinated with Cu$^2$R$_1$ (Fig. 2h). The average of the difference between the number of charges on the C atom at both ends of the C ≡ C bonds in diacetylene group changed from 1.141 to 1.447, and the results also suggested that the polarity of the diacetylene in Cu$^2$R$_1$ was also increased. As a result, the peak intensity at 2238 cm$^{-1}$ in SERS spectra of Cu$^2$R$_1$ decreased with the addition of Cu$^{2+}$.

The response time (reaching 95% of the maximum value) of CuPM@GN toward Cu$^+$ and Cu$^{2+}$ was estimated to be ~11 s and ~16 s, respectively, based on time-dependent SERS intensity ratio curves, indicating that the developed probe provided high temporal resolution (Supplementary Fig. 23). Since many other biological substances are coexisting in the intracellular environment, selectivity and competition tests of CuPM@GN for determination of Cu$^+$ and Cu$^{2+}$ against other metal ions, amino acids, common copper-containing proteins and copper-free proteins were carried out systematically. The developed CuPM@GN probe showed negligible responses (<8.9%) to potential interferences. In the competition test, no obvious changes (<9.1%) were observed after subsequent addition of these potential interferences into the solution in the presence of Cu$^{2+}$ or Cu$^+$ (Supplementary Fig. 24), indicating the good selectivity of the developed probe. Moreover, zeta potential of CuPM@GN probe was decreased from 20.8 mV to 27.9 mV, when pH values increased from 8.0 to 5.0 (Supplementary Fig. 25a), indicating that the developed CuPM@GN probe had good dispersion at pH 5.0–8.0, this may be attributed to the charge shielding effect and a certain hydrophobicity provided by the ligand layer, reducing sensitivity of the developed probe to changes in environmental pH[40]. Furthermore, it was found that the responses of the developed CuPM@GN probe toward Cu$^+$ and Cu$^{2+}$ were hardly influenced (<4.5%) when the pH value ranged from 5.0 to 8.0 (Supplementary Figs. 25b, c), and CuPM@GN signal displayed little changes (<4.3%) after the developed Raman probe was stored in cell lysis buffer for 24 h (Supplementary Fig. 26a), demonstrating the stability of the CuPM@GN probe under physiological pH conditions and its long-term sensing capability. No apparent changes (<6.6%) were found across 10 different probes, indicating excellent reproducibility (Supplementary Fig. 26b). All these results demonstrated that our developed CuPM@GN probe showed excellent accuracy, good selectivity and remarkable stability for monitoring the levels of Cu$^+$ and Cu$^{2+}$ simultaneously, which benefited from dual-recognition strategies of specific chemical reaction and characteristic Raman fingerprint peaks, as well as highly stable Au−C ≡ C bonds assembly.

## Qualifying of mitochondrial Cu$^+$ and Cu$^{2+}$ concentrations upon ischemia

For further biological application in live cells, the cytotoxicity and biocompatibility of the probe were assessed. 3-(4,5-dimethylthiazol-2yl)−2,5-diphenyltetrazolium bromide (MTT) results indicated that the cell viability remained above 90.8% even after incubating neurons with CuPM@GN (at a concentration of 0.64 mg mL$^{-1}$) for 24 h (Supplementary Fig. 27). Furthermore, little apoptotic and dead cells (<9.9%) were observed from apoptosis assay results (Supplementary Fig. 28), indicating low toxicity and excellent biocompatibility of the developed SERS probe. Meanwhile, TEM image demonstrated that CuPM@GN probes successfully entered neurons and were primarily located in the mitochondrion (Fig. 3c). Moreover, co-localization imaging experiments showed that CuPM@GN kept in mitochondria with good stability for at least 6.0 h, and the probes were then gradually expelled from the neurons within the next ~18.0 h (Fig. 3a, b). The results proved

that the developed CuPM@GN kept stable enough in mitochondria for further imaging.

Next, for SERS imaging and sensing of Cu$^+$ and Cu$^{2+}$ in live neurons, CuCl$_2$ was employed to increase the intracellular level of the copper ions[41]. As shown in Fig. 3d, with the increasing concentrations of exogenous copper ion from 0 μM to 100 μM, the pseudo color of neurons in Cu$^+$ channel changed from red-green to yellow-green, while that in Cu$^{2+}$ channel changed from red-green to orange-green. Different colors represent varied SERS ratios ($I_{2213}/I_{2155}$ and $I_{2238}/I_{2155}$) in neurons, indicating that concentrations of mitochondrial Cu$^+$ and Cu$^{2+}$ were different in neurons, and addition of exogenous copper ions resulted in both increases of mitochondrial Cu$^+$ and Cu$^{2+}$. As summarized in Fig. 3e, with increasing concentration of exogenous copper ion from 0 to 100 μM, the average signal of Cu$^+$ channel ($I_{2213}/I_{2155}$) was reduced from 3.26 ± 0.12 to 3.06 ± 0.08, while that of Cu$^{2+}$ channel ($I_{2238}/I_{2155}$) was reduced from 1.88 ± 0.09 to 1.63 ± 0.11, indicated that the concentrations of mitochondrial Cu$^+$ and Cu$^{2+}$ ($C_{Cu1}$ and $C_{Cu2}$) were increased to 1.21 ± 0.66 μM and 2.44 ± 0.56 μM, respectively. Notably, seldom copper ions were detected in the mitochondria of normal neurons, indicating the levels of intracellular free Cu$^+$ and Cu$^{2+}$ were lower than the LOD of CuPM@GN (the LOD for Cu$^+$ and Cu$^{2+}$ were 0.32 ± 0.03 μM and 0.44 ± 0.02 μM, respectively), which agrees with that previously reported results[42–44]. Interestingly, no apparent increase (<5.9%) was observed for Cu$^+$ or Cu$^{2+}$ after the neurons were incubated with CuCl$_2$ (100 μM) in the presence of bathocuproine disulfonate (BCS), a Cu$^+$ chelator, or tetrathiomolybdate (TTM), a Cu$^{2+}$ chelator. Moreover, seldom change (<3.4%) was observed for both Cu$^+$ and Cu$^{2+}$ in neurons incubated with CuCl$_2$ (100 μM) in the presence of BCS and TTM simultaneously. These results confirmed the changes in mitochondrial Cu$^+$ and Cu$^{2+}$ were induced by exogenous copper ions. These initial experiments on living neurons demonstrated that our developed CuPM@GN probe was well suited for the simultaneous detection of Cu$^+$ and Cu$^{2+}$ in mitochondria.

For further understanding of the internal mechanism of ischemia, the changes of mitochondrial Cu$^+$ and Cu$^{2+}$ in neurons were determined by using the developed CuPM@GN probe, when the neurons were treated with hypoxia and sugar deficiency (OGD), a typical cell model of ischemia[45,46]. As demonstrated in Fig. 4a, the pseudo color of both channels of Cu$^+$ and Cu$^{2+}$ were changed from red-green to orange-green after the neurons were treated with OGD, indicating apparent increase in the concentrations of mitochondrial Cu$^+$ and Cu$^{2+}$. $C_{Cu1}$ and $C_{Cu2}$ were increased to 3.09 ± 0.48 μM and 6.44 ± 0.72 μM, respectively, after the neurons were treated with OGD for 3 h (Fig. 4b, c). Interestingly, after the OGD-treated neurons were transferred to a normal culture medium for 24 h, $C_{Cu1}$ and $C_{Cu2}$ fell to 1.44 ± 0.22 μM and 2.10 ± 0.28 μM, respectively, which were obviously higher than the initial values (seldom Cu$^+$ or Cu$^{2+}$ was observed in mitochondria of neurons before treatment). The results indicated that OGD-simulated ischemia in neurons caused an irreversible increase in mitochondrial copper concentrations.

Previous literatures have reported that intracellular copper ions were almost bound and stored by various metal-binding proteins rather than free copper ions to maintain copper homeostasis and avoid free copper ions-involved oxidative stress damage, especially in mitochondria, due to mitochondria are important copper reservoirs in cells and physiological functions of mitochondria are strictly regulated by copper ions and oxidative stress[47,48]. However, we found a remarkable accumulation of mitochondrial copper ions during ischemia. To understand the reason for the increase in Cu$^{2+}$ concentration (~6.44 μM) higher than that of Cu$^+$ (~3.09 μM) during OGD treatment, two possible pathways for the changes in the concentrations of copper ions were further investigated, including the copper ions released from copper-containing proteins as well as the conversion between Cu$^+$ and Cu$^{2+}$. Firstly, the levels of cytochrome C oxidase 17 (Cox 17) and synthesis of cytochrome c oxidase 1 (SCO1) proteins were examined

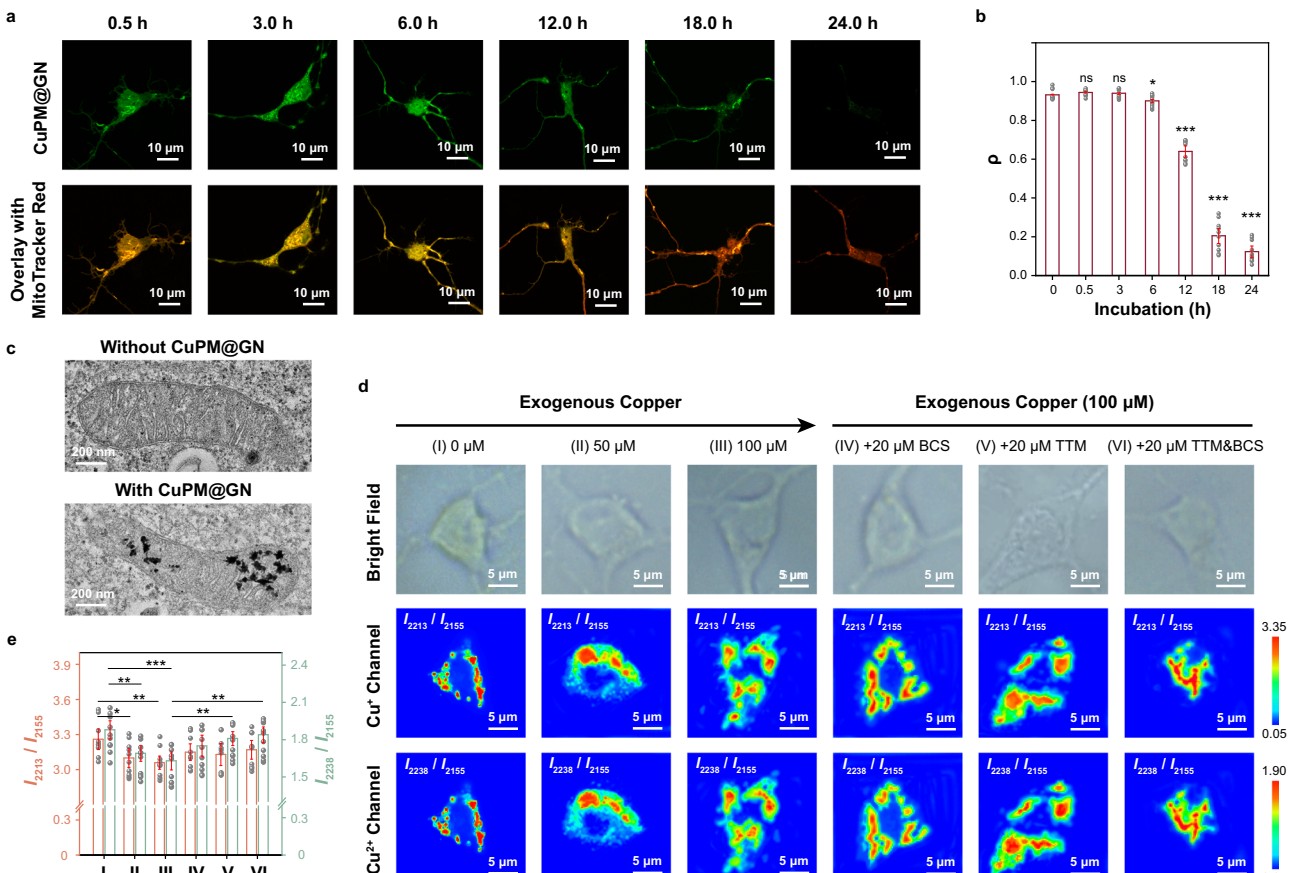

**Fig. 3 | Simultaneously quantifying of Cu⁺ and Cu²⁺ in mitochondria. a** Typical co-localization images of neurons incubated with CuPM@GN and MitoTracker Red for different incubation times. Three independent experiments were repeated and similar results were obtained. **b** Pearson's correlation coefficients ($\rho$) of CuPM@GN with mitochondria corresponding to (**a**) ($n = 10$ cells). **c** Representative TEM image of CuPM@GN probes trapped in a mitochondrion. Three independent experiments were repeated and similar results were obtained. **d** Raman scanning mapping images obtained from neurons treated with 0 µM (I), 50 µM (II) and 100 µM CuCl₂ (III), and treated with 100 µM CuCl₂ in the presence of 20 µM BCS (IV), 20 µM TTM (V) and 20 µM BCS + 20 µM TTM (VI), respectively. **e** The SERS intensity ratios of $I_{2213}/I_{2155}$ and $I_{2238}/I_{2155}$ in neurons corresponding to panel (d) ($n = 10$ cells). The above-mentioned data are all presented as mean ± S.D. Error bars: S.D., gray dots represent individual data points. Statistical significance is calculated with a two-tailed unpaired $t$-test and $P$ values are indicated ($^{ns}P > 0.05$, $^{*}P \leq 0.05$, $^{**}P \leq 0.01$, $^{***}P \leq 0.001$). Source data are provided as a Source Data file.

before and after the neurons were treated with OGD, because Cox 17 and SCO1 were typical Cu⁺ and Cu²⁺ chaperone proteins responsible for Cu⁺ and Cu²⁺ transport and storage in cells[49]. As the western blot (WB) results shown in Fig. 4d, the levels of Cox 17 (Cox 17/β-actin) and SCO1 (SCO1/β-actin) in normal neurons were estimated as 0.46 ± 0.03 and 0.19 ± 0.03, by using β-actin as reference standard protein. However, after the neurons were treated with OGD for 3 h, Cox 17/β-actin and SCO1/β-actin reduced to 0.25 ± 0.04 and 0.11 ± 0.05, respectively (Fig. 4e). These results indicated that ~45.65% of Cox 17 and ~33.33% of SCO1 were destroyed during ischemia. Therefore, the structural destruction of the copper-containing proteins (such as Cox 17 and SCO1) was considered as important source for the elevated levels of free copper ions in neurons during ischemia.

On the other hand, the possible conversion between mitochondrial Cu⁺ and Cu²⁺ in neurons during OGD was also investigated. As shown in Fig. 4a–c, the level of free Cu²⁺ only increased to ~3.77 µM, after the neurons were treated with OGD in the presence of BCS, while seldom increase was observed for Cu⁺ (<0.32 ± 0.03 µM, which was the LOD for Cu⁺). The results strongly proved that the increased Cu²⁺ partly came from Cu⁺ conversion during OGD treatment. Our previous work in vivo proved that the concentration of extracellular Cu²⁺ was reduced by ~66.00% and that of Cu⁺ concentration was increased by ~36.00% in the cerebral cortex after the mouse was treated with glutathione (GSH)

during ischemia[23], therefore, the effect of GSH on mitochondrial copper ions during OGD-simulated ischemia was also studied (Fig. 4a), because GSH is one of the most potent natural scavengers of ROS[50,51]. With the concentrations of pre-incubated GSH increased from 0.1 mM to 1.0 mM, $C_{Cu2}$ reduced by ~30.20% (to 4.49 ± 0.56 µM) and ~68.63% (to 2.02 ± 0.37 µM), compared with $C_{Cu2}$ in ischemic neurons in the absence of GSH. To our surprise, $C_{Cu1}$ increased by ~14.33% (to 3.53 ± 0.49 µM) and ~21.54% (to 3.75 ± 0.41 µM) under the same experimental conditions, respectively. The results proved that reduction of ROS during ischemia inhibited Cu²⁺ increase and promoted Cu⁺ increase in mitochondria. Since ROS is an important participant in the conversion of Cu⁺ and Cu²⁺[52], these results further suggested that the conversion of Cu⁺ to Cu²⁺ is also a critical route for the increasing of mitochondrial copper ions during ischemia, especially for the increasing of Cu²⁺.

Interestingly, cell viability of neurons decreased with the prolongation of the OGD time, and the cell viability of neurons decreased from 98.1% to 30.9% after the neurons were treated with OGD for 3 h (Fig. 4 f), but addition of GSH (1.0 mM) caused the viability of neurons significantly recovered to 74.6%, demonstrating that inhibiting the conversion of Cu⁺ to Cu²⁺ effectively protected neurons from oxidative stress. Moreover, after neurons were treated with OGD in the presence of MK-801, a widely used antagonist that could inhibit the transport of

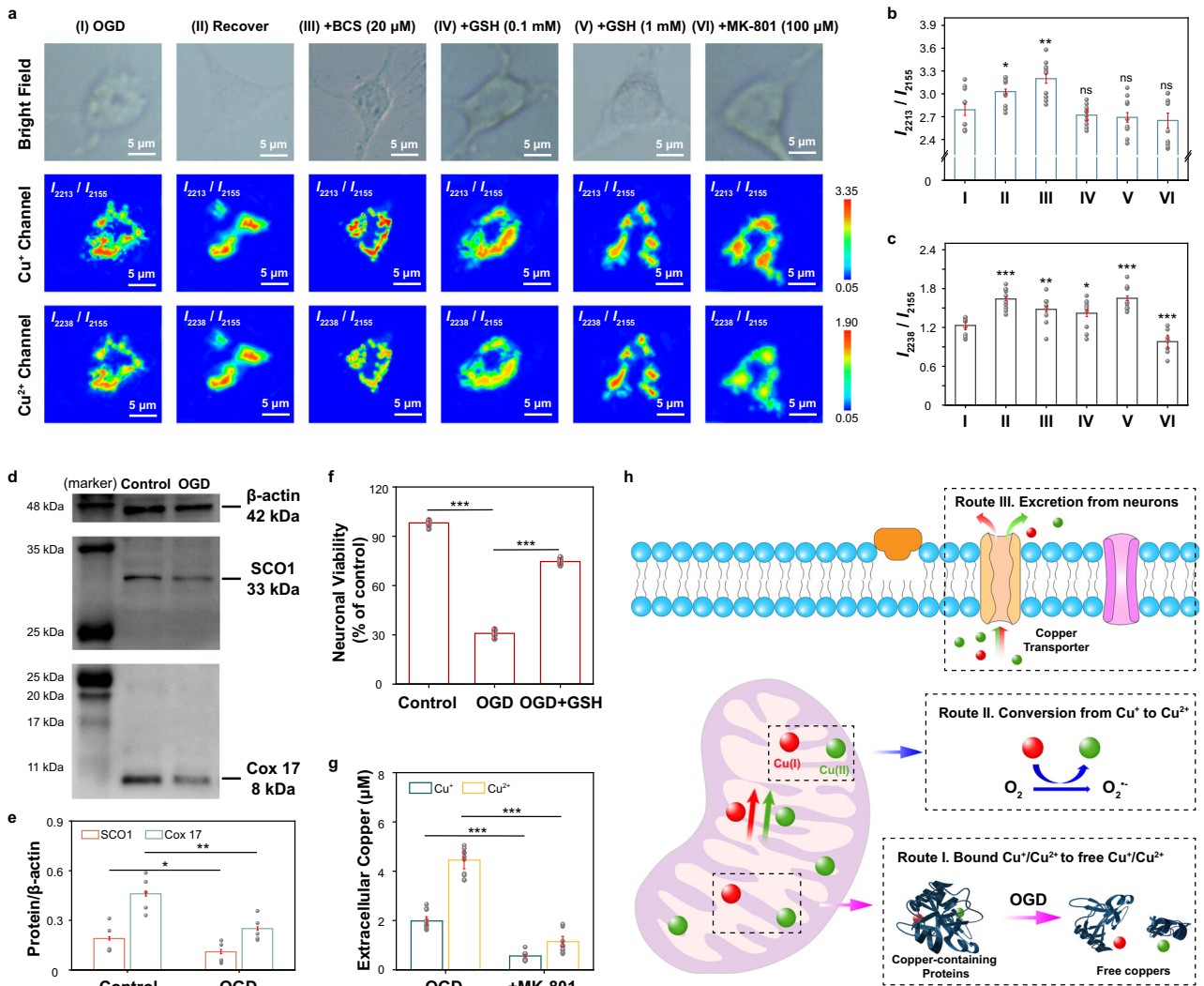

**Fig. 4 | Imaging and biosensing of mitochondrial Cu$^+$ and Cu$^{2+}$ concentrations upon ischemia. a** Raman scanning mapping images obtained from neurons pre-treated with BCS (20 μM), GSH (0.1 mM and 1 mM) and MK-801 (100 μM) for 12 h and then treated with oxygen-glucose deprivation (OGD) for 3 h. For comparison, neurons were treated with OGD for 3 h and then transferred to a normal medium. **b**–**c** The SERS intensity ratios of $I_{2213}/I_{2155}$ (**b**) and $I_{2238}/I_{2155}$ (**c**) corresponding to panel **a** ($n = 10$ cells). **d** Western blot images of proteins (Cox 17, SCO1 and β-actin) obtained from neurons before and after OGD treatment. All samples were derived from the same experiment and the blots were processed in parallel. Uncropped blots in Source Data. **e** The levels of Cox 17/β-actin and SCO1/β-actin corresponding to (**d**) ($n = 5$ independent experiments). **f** The viability of neurons before (control) and after OGD treatment for 3 h (I), and OGD-treated neurons pre-treated with 1 mM GSH (II), respectively ($n = 10$ independent experiments). **g** Extracellular Cu$^+$ and Cu$^{2+}$ levels in the culture medium of OGD-treated neurons with and without MK-801 pre-treatment ($n = 10$ independent experiments). **h** Schematic illustration for routes for understanding the increases of mitochondrial Cu$^+$ and Cu$^{2+}$ upon ischemia. The above-mentioned data are all presented as mean ± S.D. Error bars: S.D., gray dots represent individual data points. Statistical significance is calculated with a two-tailed unpaired $t$-test and $P$ values are indicated (ns$P > 0.05$, *$P \leq 0.05$, **$P \leq 0.01$, ***$P \leq 0.001$). Source data are provided as a Source Data file.

copper ions to the outside of the cell through ATPase[53], $C_{Cu1}$ and $C_{Cu2}$ were increased to $4.03 \pm 0.66$ μM and $9.07 \pm 1.07$ μM, respectively (Fig. 4a–c), which were elevated by ~30.42% and ~40.78%, compared with the results obtained from the ischemic neurons in the absence of MK-801. Moreover, by using our previously developed probe[23], we found that the concentrations of extracellular Cu$^+$ and Cu$^{2+}$ in the culture medium were decreased by ~71.72% and ~74.42% after the neurons treated with OGD in the presence of MK-801 (Fig. 4g). These results together proved that the increased copper ions in neurons were partially excreted to the outside of the cell by ATPase during ischemia (Fig. 4h), which supported our previous hypothesis at the in vivo level[23]. All these findings benefited from our developed SERS probe for the determination of Cu$^+$ and Cu$^{2+}$ with high selectivity and high accuracy.

## The effect of Cu$^+$ to Cu$^{2+}$ conversion on neuron death processes in cuproptosis

The mechanism of cell death induced by copper ions overload is unclear until proteotoxic stress-induced cuproptosis was discovered[26]. The finding was based on the cell viability in hundreds of cell lines after the cells were stimulated by overloaded Cu$^{2+}$ for ~24 h. To our surprise, copper-induced cytotoxicity was also observed in neurons and other nerve model cells (PC-12, SH-SY5Y and HT22) (Fig. 5a), after the cells were stimulated by CuCl$_2$ in the presence of elesclomol (ES), a copper ionophore (ES-Cu)[54]. In addition, no obvious (<8.2%) neuron viability was observed after the neurons were stimulated by ES in the presence of other metal ions, including iron and zinc ions (Supplementary Fig. 29a). Moreover, the neuron viability gradually decreased with increasing concentration of ES-Cu as well as prolonging stimulation

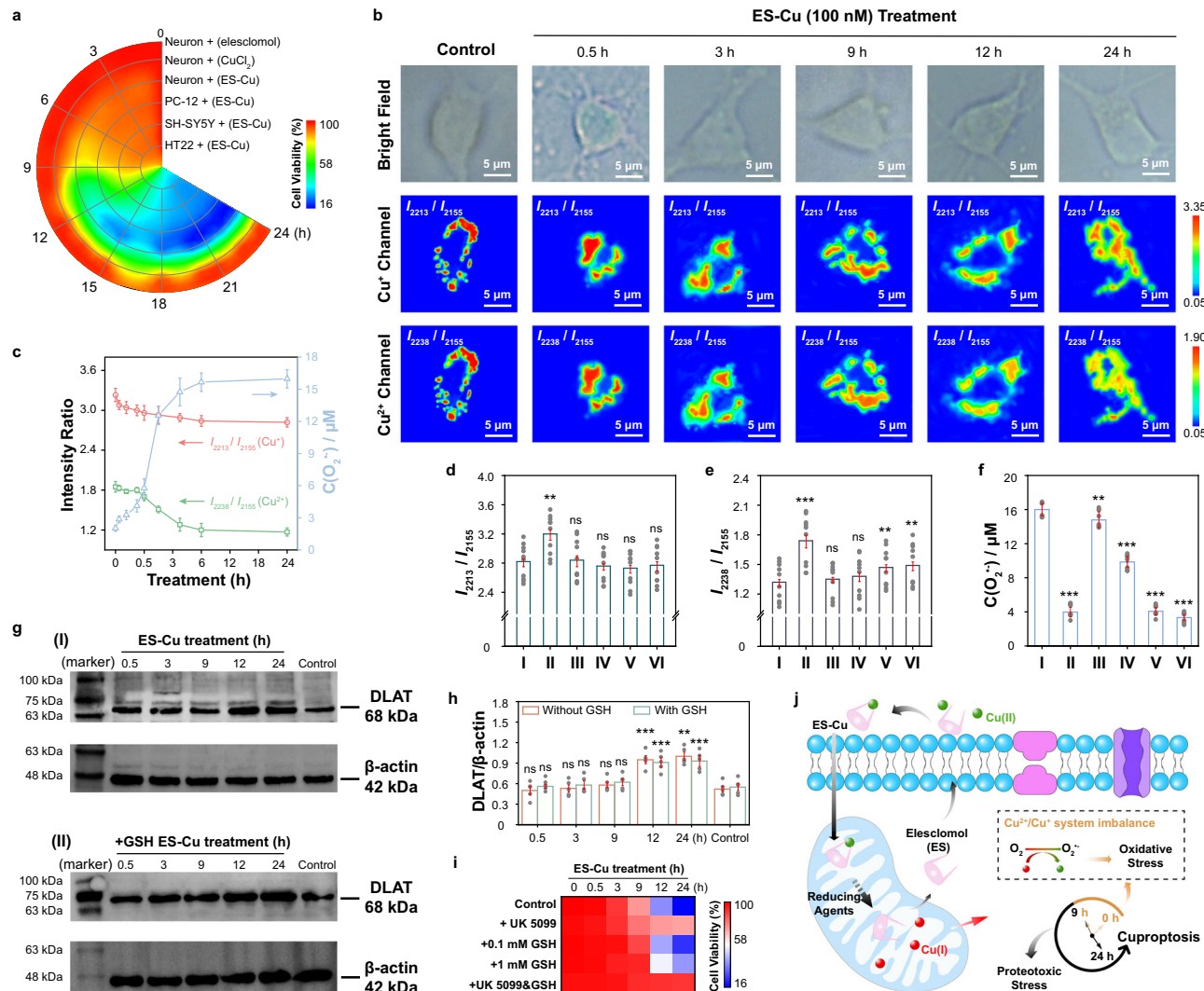

**Fig. 5 | The effect of Cu⁺ to Cu²⁺ conversion on neuronal death caused by copper overload. a** Time-dependent cell viability measurements of 4 cell lines treated with 100 nM elesclomol-CuCl₂ (ES-Cu,1:1) and neurons treated with 100 nM elesclomol and 100 nM CuCl₂, respectively. **b** Raman mapping images obtained from neurons during 100 nM ES-Cu-induced copper overload at different time points. **c** The concentration of O₂˙⁻ (C(O₂˙⁻)) and the SERS intensity ratios of $I_{2213}/I_{2155}$ and $I_{2238}/I_{2155}$ obtained from mitochondria in neurons at different time points upon 100 nM ES-Cu-induced copper overload (n = 10 cells). **d–f** $I_{2213}/I_{2155}$ (**d**), $I_{2238}/I_{2155}$ (**e**) and C(O₂˙⁻) (**f**) obtained from mitochondria treated with 100 nM ES-Cu (I) for 24 h, pre-treated with 20 μM BCS (II), 100 nM UK 5099 (III), 0.1 mM GSH (IV), 1 mM GSH (V), as well as the mixture of 1 mM GSH and 100 nM UK 5099 (VI) (n = 10 cells). **g** Western blot images of proteins (DLAT and β-actin) obtained from neurons stimulated by ES-Cu

complex at different intervals in the absence (I) and presence (II) of 1.0 mM GSH. All samples were derived from the same experiment and the blots were processed in parallel. Uncropped blots in Source Data. **h** The levels of DLAT/β-actin corresponding to (**g**) (n = 5 independent experiments). **i** Time-dependent cell viability measurement of neurons stimulated by 100 nM ES-Cu upon different pre-processed, respectively. **j** Schematic diagram of the mechanism of cell death induced by copper overload. The above-mentioned data are all presented as mean ± S.D. Error bars: S.D., gray dots represent individual data points. Statistical significance is calculated with a two-tailed unpaired t-test and P values are indicated (ⁿˢP > 0.05, *P ≤ 0.05, **P ≤ 0.01, ***P ≤ 0.001). Source data are provided as a Source Data file.

time (Supplementary Fig. 29b). These results indicated cuproptosis also occurred in neurons during copper overload.

To understand the exact roles of Cu⁺ and Cu²⁺ in neuronal death caused by copper overload, the dynamic changes of Cu⁺ and Cu²⁺ in mitochondria were quantified by our developed SERS probe after being stimulated by overload copper ions. As shown in Fig. 5b, both the concentrations of mitochondrial Cu⁺ and Cu²⁺ were gradually increased after the neurons were stimulated by ES-Cu. Interestingly, the concentration of Cu⁺ in mitochondria (C_Cu1) rapidly increased from a very low level (<0.32 ± 0.03 μM, which was the LOD for Cu⁺) to 1.01 ± 0.31 μM within the initial 0.5 h following the stimulation of extracellular ES-Cu complex (Fig. 5c). This observation was attributed to the passive diffusion of ES-Cu complex into neurons and released free Cu⁺ upon the action of intracellular reducing agents[55,56].

Meanwhile, the concentration of mitochondrial Cu²⁺ (C_Cu2) also increased from very low level (<0.44 ± 0.02 μM, which was the LOD for Cu²⁺) to 0.66 ± 0.10 μM within 1.0 h following the changes of mitochondrial Cu⁺. Surprisingly, C_Cu1 and C_Cu2 were increased to ~2.35 ± 0.43 μM and ~5.89 ± 1.04 μM after the neurons were stimulated by ES-Cu complex for 9 h, and kept stable in the following ~15 h. Interestingly, the level of free Cu²⁺ only increased to ~1.06 μM, after the neurons were treated with ES-Cu complex for 9 h in the presence of BCS, while seldom increase was observed for Cu⁺ (<0.32 ± 0.03 μM, which was the LOD for Cu⁺) (Fig. 5d, e), indicating that part of the released free Cu⁺ was then converted into Cu²⁺ during the process. The results demonstrated that both Cu⁺ and Cu²⁺ were obviously increased in mitochondria, especially in the early stages (<9 h) of ES-Cu complex stimulation.

On the other hand, it was found that the concentration of super-oxide anion ($O_2^{\cdot-}$) in mitochondria was also gradually increased from $2.02 \pm 0.28\ \mu M$ to $14.75 \pm 1.30\ \mu M$ after the neurons were stimulated by ES-Cu complex for 9 h (Fig. 5c), while seldom increase (<8.34%) was observed for $O_2^{\cdot-}$ in mitochondria with the prolongation of stimulation time (>9 h), by using our previously developed $O_2^{\cdot-}$ probe[57]. Interestingly, it was observed that the concentration of $O_2^{\cdot-}$ in mitochondria was decreased to $3.97 \pm 0.71\ \mu M$ after the neurons were treated with ES-Cu complex for 9 h in the presence of BCS (Fig. 5 f). The results confirmed that the conversion of $Cu^+$ to $Cu^{2+}$ occurred during copper overload, which was accompanied by $O_2^{\cdot-}$ burst simultaneously. It should be pointed out that the concentration of $O_2^{\cdot-}$ in mitochondria ($3.97 \pm 0.71\ \mu M$) after the neurons were treated with ES-Cu complex in the presence of BCS was still higher than that in neurons without 100 nM ES-Cu-treated ($2.02 \pm 0.28\ \mu M$), suggesting that other processes may also contribute to the increase of mitochondrial $O_2^{\cdot-}$ during copper overload, such as the leakiness of the electron transport chain (ETC) and the abnormal synthesis of ATP[58].

To prevent the conversion of $Cu^+$ to $Cu^{2+}$, different concentrations of GSH (0.1 mM and 1.0 mM) were used to scavenge ROS in cells[59]. As shown in Fig. 5d, e, after the neurons were stimulated by overload copper ions for 24 h in the presence of a low concentration of GSH (0.1 mM), $C_{Cu2}$ reduced by ~17.20% (to $4.90 \pm 0.58\ \mu M$), while $C_{Cu1}$ increased by ~15.13% (to $3.24 \pm 0.37\ \mu M$), compared with those of the copper overload neurons without GSH. Meanwhile, the concentration of $O_2^{\cdot-}$ in mitochondria was decreased by ~38.48% (to $9.85 \pm 0.97\ \mu M$) (Fig. 5f). More interestingly, with the concentration of the stimulated GSH increased to 1.0 mM, $C_{Cu2}$ further reduced to $3.95 \pm 0.49$ (~33.26%), while $C_{Cu1}$ increased to $3.44 \pm 0.44\ \mu M$ (~22.14%) (Supplementary Fig. 30). In addition, the concentration of mitochondrial $O_2^{\cdot-}$ was further decreased by ~74.64% (to $4.06 \pm 0.71\ \mu M$). The results strongly indicated that the conversion of $Cu^+$ to $Cu^{2+}$ is critical for the increase in $O_2^{\cdot-}$ concentration during copper overload. The more mitochondrial $Cu^{2+}$ in the oxidized state and the less $Cu^+$ in the reduced state, the greater the concentration of $O_2^{\cdot-}$ in mitochondria, resulting in significant oxidative stress.

Moreover, no obvious change (<12.73%) was observed from WB results for the levels of DLAT after the neurons were stimulated by ES-Cu complex for 0.5 h, 3 h or 9 h, compared with those obtained with no stimulation of ES-Cu complex (Fig. 5g, h). In contrast, the levels of DLAT were increased by ~82.69% and ~92.31% after the neurons were stimulated by ES-Cu for 12 h and 24 h, respectively, indicating that proteotoxic stress was the dominant event at later stage (>9 h) of cuproptosis. To our surprise, the levels of DLAT decreased by ~17.24% and ~23.22% after the neurons were stimulated by ES-Cu for 12 h and 24 h in the presence of GSH (1.0 mM), respectively, while seldom change was observed for those levels of DLAT within 9 h (Fig. 5g, h), proving the level of proteotoxic stress at later stage (>9 h) of cuproptosis was reduced by suppressing oxidative stress. All these results indicated that oxidative stress ($O_2^{\cdot-}$ burst) caused by the conversion of $Cu^+$ to $Cu^{2+}$ in mitochondria was the leading event at the early stage (<9 h) of cuproptosis, which partially exacerbated the proteotoxic stress that dominated the later stage (>9 h) of cuproptosis.

Time-dependent cell viability results showed that the cell viability gradually decreased with the prolongation of stimulation time by ES-Cu complex. Interestingly, cell viability obviously recovered to ~74.8% (from ~39.1%) and ~73.2% (from ~16.2%) after being stimulated by ES-Cu complex in the presence of cuproptosis inhibitor (UK 5099, 100 nM) for 12 h and 24 h, respectively. Meanwhile, no apparent improvement (<5.7%) was observed for neurons under the same stimulation within 9 h. The results demonstrated that proteotoxic stress was the main cause of neuron death in the later stage (>9 h) of cuproptosis, addition of cuproptosis inhibitor had no obvious effect on oxidative stress-induced neuronal death at the early stage (<9 h) of cuproptosis. However, with the addition of 0.1 mM and 1.0 mM GSH, the viability of

neurons was recovered from ~75.1% to ~88.3% and ~94.0% after the neurons were stimulated by ES-Cu complex for 9 h, proving that oxidative stress was the main cause of neuron death in the early stage (<9 h) of cuproptosis. More importantly, after 24 h of stimulation, the viability of neurons increased from ~16.2% to ~23.9% and ~40.6% in the presence of 0.1 mM and 1.0 mM GSH, respectively, further evidencing that oxidative stress worsened proteotoxic stress-induced neuronal death, which may be due to the exacerbation of proteotoxic stress caused by oxidative stress during cuproptosis. To our surprise, seldom neuronal death (<8.8%) was observed even after the neurons were stimulated by ES-Cu complex for 12 or 24 h in the presence of UK 5099 (100 nM) and GSH (1.0 mM) simultaneously. These results together proved that both oxidative stress and proteotoxic stress were involved in cuproptosis. In particular, oxidative stress primarily contributed to neuron death at the early stage (<9 h) of cuproptosis, which exacerbated proteotoxic stress-dominated neuron death at the later stage (> 9 h) of cuproptosis. The finding benefited from the developed SERS probe for real-time sensing and quantitative determination of mitochondrial $Cu^+$ to $Cu^{2+}$ simultaneously.

## Discussion

In this work, a SERS probe (CuPM@GN) was developed for simultaneously imaging and biosensing of $Cu^+$ and $Cu^{2+}$ in mitochondria of neurons, based on the design of specific chemical reactions toward $Cu^+$ and $Cu^{2+}$, resulting in well-separated characteristic Raman fingerprint peaks. A series of molecules containing different diacetylene derivatives as Raman reporters were designed, whose Raman signals were located in the cellular Raman-silent region and effectively avoided potential interference from biological species. Combined with theoretical calculations, two molecules with the best Raman signals, $Cu^1R_5$ and $Cu^2R_1$, were optimized for the determination of $Cu^+$ and $Cu^{2+}$, respectively. Using our developed tool, the level of free copper ions in mitochondria, especially $Cu^{2+}$, was found significantly increased under hypoxia and sugar deficiency-simulated ischemia. Two possible pathways were found to explain the increases of $Cu^+$ and $Cu^{2+}$ during ischemia: the structurally damaged copper-containing proteins and the conversion of $Cu^+$ to $Cu^{2+}$ during ischemia. In addition, the increased free copper ions were partially excreted from neurons through ATPase during ischemia was also discovered. Moreover, both oxidative stress and proteotoxic stress contributed to neuron death during cuproptosis. Interestingly, we discovered that oxidative stress caused by the conversion of $Cu^+$ to $Cu^{2+}$ in mitochondria was the leading event at the early stage (<9 h) of cuproptosis, which partially exacerbated proteotoxic stress that dominated the later stage of cuproptosis.

Our work developed a way for simultaneous assessment of mitochondrial $Cu^+$ and $Cu^{2+}$ levels in live neurons, which facilitates the investigation of molecular mechanisms underlying copper accumulation-related physiological and pathological processes within mitochondria. In addition, this work has provided a methodology for not only demonstrating different pathways for copper increases during ischemia, but also providing insights into oxidative stress-induced neuronal injury and copper overload-induced cuproptosis. Due to the wide shift distribution at the cellular Raman-silent region (1800-2600 cm$^{-1}$) and Raman fingerprint peak with a narrow half-width, more related species can be further detected in neurons or even different organelles at the same time, for eventually understanding the molecular interactions and networks related to neurological diseases, including hypoxia, ischemia, epilepsy and stroke.

## Methods
### Materials

All chemicals were obtained from commercial suppliers (unless otherwise specified), meanwhile, analytical grade reagents and solvents were procured from Sinopharm Chemical Reagent Co. Ltd.

(Shanghai, China). All purchased chemicals can be used without purification. DNA oligonucleotides modified (5′-HC≡C-(T)$_{15}$ AGCG-CAGGCC-(6-FAM)−3′) were ordered and purified from Sangon Biotech Co., Ltd (Shanghai, China). Reagents related to neuron culture, including B27, trypsin, and neurobasal medium, were obtained from Thermo Fisher Scientific (U.S.A.). Other used cell lines were purchased from commercial companies (Enzyme-linked Biotechnology Co., Ltd. Shanghai), including HT22 (ml096819), SH-SY5Y (ml097536), and PC-12 (ml096373). Cu$^+$ solution dissolved in acetonitrile was prepared from [Cu (CH$_3$CN)$_4$][PF$_6$] under a nitrogen atmosphere. Cu$^{2+}$ solution dissolved in water was obtained from CuCl$_2$·2H$_2$O under nitrogen protection.

## Synthesis

The synthetic procedures and details of the compounds mentioned in this report can be found in the Supplementary Information file.

## Apparatus and Instruments

NMR spectra and mass spectra of molecules were obtained from an AC-80 NMR spectrometer (BrukerBioSpin, Germany) and a micrOTOF II mass spectrometer (Bruker, Germany). UV-vis absorption spectra were measured on a Hitachi 5300 spectrophotometer (Hitachi High-Technologies Co., Japan), respectively. TEM images were obtained from a Talos F200C G2 transmission electron microscope (Thermo Fisher Scientific, U.S.A.). A Zetasizer Nano (Malvern Instruments, Ltd.) was used to perform dynamic light scattering (DLS) measurements. A FACS Calibur flow cytometer (BD Biosciences, U.S.A.) was used to conduct cell apoptosis assay experiments. TCS-SP8 confocal laser-scanning microscope (Leica, Germany) was used to obtain confocal fluorescence images.

The SERS measurements and SERS imaging were conducted on inVia Raman microscope (Renishaw, U.K.). A 785 nm laser with a maximum (100%) laser power of 300 mW was used for all the measurements. Baseline subtraction (cubic spline interpolation) and smoothing (Savitzky–Golay, polynomial order 3, smooth window 9) of Raman Spectra were performed using Renishaw WiRE 5.1 software. For SERS measurements of solution samples, a 10× microscope objective (N.A. = 0.25, working distance: 17.6 mm) and a spot-focused laser were utilized. The laser power of 30 mW (10%) and acquisition time of 5 s were selected. When conducting SERS imaging of live cells, the imaging data were collected using the Raman mapping mode, employing a 50× microscope objective (N.A. = 0.5, working distance: 8.2 mm) and a spot-focused laser. The laser power of 3.0 mW (1%) and acquisition time of 1 s were selected. Finally, the ratio values of each data point collected in the Raman scanning mapping images are mapped to the grayscale values to obtain visual pseudo-color images, which are used to reflect the overall distribution. The relationship between color span and value range is shown in each diagram.

## Calculation of binding constant

The binding constant $K$ of Cu$^1$R$_5$/Cu$^+$ complex and Cu$^2$R$_1$/Cu$^{2+}$ complex were calculated according to the following Benesi-Hildebrand equation[60]:

$$\frac{1}{A - A_0} = \frac{1}{A_{max} - A_0} + \frac{1}{K[M]} \times \frac{1}{A_{max} - A_0} \quad (1)$$

Where $K$ was the binding constant, $A_0$ and $A$ represented the observed absorption in the absence and the presence of ions, respectively. $A_{max}$ was the observed absorption when ions reached saturation and $[M]$ was the concentration of added ions.

## Preparation of CuPM@GN

Gold nanostars (GNs) were synthesized by the seed growth method[61]. [4-oxo-4-(2-propyn-1-ylamino) butyl] triphenylphosphonium (TPP),

1-ethynyl-4-[2-[tris(1-methylethyl) silyl] ethynyl] benzene (EETP), Cu$^1$R$_5$ and Cu$^2$R$_1$ dissolved into DMSO mixed solution (100: 25: 100: 100 μM) was added to the aqueous solution of GNs (1.00 mg mL$^{-1}$). Next, N$_2$ was bubbled into the above solution for 0.5 h to expel the dissolved O$_2$, and then the solution was stirred at 60 °C for 2 h under N$_2$ atmosphere. Eventually, the solution underwent centrifugation for 10 min at 3000 g, after which the supernatant was discarded to eliminate excess molecules from the solution. After repeating the centrifugation 3 times, the nanoparticles were resuspended in 20 mM PBS (pH = 7.4) to obtain SERS probe (CuPM@GN).

## Raman enhancement factor calculation

Raman enhancement factor (EF) was calculated as follows:

$$EF = \frac{I_{SERS} N_{bulk}}{I_{bulk} N_{ads}} \quad (2)$$

Where $N_{ads}$ and $N_{bulk}$ represent the number of EETP molecules in the SERS sample (GNs) and the normal Raman sample, respectively. $I_{SERS}$ and $I_{bulk}$ were the same vibration peaks (1595 cm$^{-1}$) of Raman spectra generated from $N_{ads}$ and $N_{bulk}$, respectively.

## Cell culture

All animal and cell experiments followed the guidelines outlined in the Care and Use of Laboratory Animals by the Ministry of Science and Technology of China. Approval for these experiments was granted by the Animal Care and Use Committee of East China Normal University (approval no. m + R20190304, Shanghai, China). The neurons used in the experiment were all derived from the cerebral cortex of neonatal C57BL/6 mice[62]. Neonatal mice (within 24 h) were anesthetized with isoflurane, and their entire brains were promptly extracted and immersed in Hank's balanced salt solution (HBSS) lacking Ca$^{2+}$ and Mg$^{2+}$ at 0 °C. Then, the cortical tissue was dissected under a microscope and co-incubated with papain at 37 °C for 15 mins. The neurons were re-suspended in a small amount of plating medium, and then plated in poly-D-lysine-coated (PDL) culture dishes. After incubation for 4 h in the cell culture incubator, the plating medium was replaced with freshly prepared growth medium (48.0 mL neurobasal culture medium + 1.0 mL 0.5 M B27 + 0.5 mL 1 M P/S + 0.5 mL 1 M L-glutamine). The dishes were returned to the incubator for continued culture (CO$_2$: 5%; air: 95%; 37 °C), and the growth medium was refreshed every three days. After 5–7 days, the neurons grew sufficiently for subsequent experiments.

SERS imaging was performed using a confocal Raman spectrometer. Initially, neurons were exposed to a medium containing the SERS probe CuPM@GN (0.46 mg mL$^{-1}$) for 0.5 h within the incubator. After that, removing the culture medium and the neurons were rinsed three times with PBS to remove any nanoparticles that were not absorbed and remained external to the cells. Finally, the petri dish was subjected to observation via the objective lens of the confocal Raman spectrometer for SERS imaging experiments.

## Stability of CuPM@GN

To study the stability of the developed CuPM@GN probe in mitochondria, 6-FAM labeled DNA strand was further conjugated onto CuPM@GN probe as fluorescent indicator. 6-FAM labeled CuPM@GN probe (6-FAM-CuPM@GN) was only used to estimate the stability of CuPM@GN probe by fluorescence co-localization imaging, and not used for SERS measurements. Neurons were incubated with medium containing 6-FAM-CuPM@GN (0.46 mg mL$^{-1}$) for different time intervals (0.5, 3, 6, 12, 18, and 24 h) in the incubator and further used for fluorescent imaging. The fluorescent signal was collected between 500-540 nm under the excitation of 488 nm.

## Cytotoxicity and apoptosis assays

Neurons were pre-incubated in 96-well plates with various CuPM@GN (0, 0.18, 0.32, 0.46, 0.50 and 0.64 mg mL$^{-1}$) and then were cultured for 24 h for cytotoxicity assessment. Then, MTT (20 µL) solution was added subsequently to each well, and then the mixtures were further incubated for another 4 h in darkness. After removing the mixed solution, DMSO (80 µL) was added to each well of 96-well plates and further shaken for 5 min. The absorbance at 490 nm in each well of 96-well plates was measured by a Varioskan LUX microplate reader (Thermo Fisher Scientific). Finally, cell viabilities (%) were assessed by calculating the ratio of absorbance in the experimental group to that in the blank control group.

As for apoptosis analysis, neurons were pre-treated with various concentrations of CuPM@GN (0, 0.18, 0.46 and 0.64 mg mL$^{-1}$) for 24 h. Then, the medium was removed, cells were detached using EDTA-free trypsin, washed with HBSS, and then re-suspended in 300 µL of binding buffer. The cell suspension was then added with both 5 µL of propidium iodide (PI) solution and 5 µL of FITC-Annexin V for a 30-minute incubation in darkness. Finally, flow cytometry results analysis was conducted using FlowJo (X 10.0.7 R2).

## Western Blot analysis

Mitochondria Isolation Kit was obtained from KeyGEN BioTECH (KGB5401, Jiangsu, China). The culture mediums of each group were discarded and then washed with cold PBS twice. After adding the cell lysate, the cells were scraped away on the ice, and the cell lysate containing neurons was transferred to the PE tubes. After shaking for 30 min, the solution was centrifuged at 800 g (5 min, 4 °C). The upper layer of Medium Buffer in centrifuge tube was covered with obtained supernatant (v/v = 1:1, 4 °C), and then centrifuged at 15000 g (10 min, 4 °C). After resuspending the precipitate with Wash Buffer, the centrifugation was repeated at 15000 g (10 min, 4 °C), and the precipitate obtained was the extracted mitochondria. Next, the protein was extracted by the kit, which was obtained from Solarbio Science & Technology Co., Ltd. (Beijing, China, EX1321), and the liquid supernatant collected from mitochondrial precipitates was stored at −80 °C. Subsequently, the concentrations of protein in the supernatants were measured using the BCA protein assay kit obtained from Beijing Solarbio Science & Technology Co., Ltd. (Beijing, China, PC0020) with a standard of bovine serum albumin. The mixed solution containing the sample, PBS, and loading buffer was boiled and then loaded onto a 4–20% polyacrylamide gel for electrophoresis. Afterward, the gel was transferred onto a PVDF membrane, and then the membrane was blocked with a blocking buffer for 1 h (25 °C). After that, the membrane further was incubated respectively with the primary antibody against cytochrome C oxidase 17 (Cox 17, 1: 2000, Abcam ab69611), primary antibody against synthesis of cytochrome c oxidase 1 (SCO1, 1:2000, OriGene Technologies, TA381301), primary antibody against dihydrolipoamide S-acetyltransferase (DLAT, 1:1000, OriGene Technologies, TA350579), and primary antibody against β-actin (1:5000, Abcam ab8226) overnight at 4 °C. Then the membranes were incubated with peroxidase-conjugated goat anti-rabbit IgG (H + L) (1: 10000, YEASEN 33119ES60) for 1 h at room temperature. Finally, the membranes were incubated with an electrochemiluminescence (ECL) solution and then imaged on a ChemiDoc Touch imaging system.

## Statistics and reproducibility

The times of some experiments repeated independently with similar results are stated in the legends. For other experiments (such as micrographs), three independent experiments are conducted, yielding similar results. Data are expressed as mean values ± standard deviation (S.D.), calculated using Microsoft Excel 2016. Statistical analysis is performed with an unpaired two-tailed Student's t-test using IBM SPSS 27 statistical software, followed by post hoc tests for multiple comparisons. $P$ value < 0.05 is considered statistically significant ($^{ns}P > 0.05$, $*P \leq 0.05$, $**P \leq 0.01$, $***P \leq 0.001$). Exact $P$ values are provided in each figure legend. Other test results such as confidence intervals are provided in the Source Data file. The $n$ values, as indicated within the figure legends, represent biologically independent samples or independent experiments. Derived statistics are based on the analysis of averaged values across biological replicates, rather than technical replicates.

## Theoretical calculation

The Gaussian 09 suite of programs[63] was used to optimize the configurations. Structural optimization and electrostatic potential surfaces were performed by the b3lyp functional. The 6–31 G (d) basis set was employed for the C, H, O, N, and S atoms, and the SDD basis set was used for the Cu atom. The maximum φ ($\varphi_{max}$) and minimum φ ($\varphi_{min}$) of the four materials were obtained with the help of Multiwfn code[64]. The analysis of electrostatic potential (ESP) was performed by a multifunctional wavefunction analysis program Multiwfn 3.4.1[64]. All iso-surface maps were generated using the VMD 1.9.1 program[65], utilizing the outputs obtained from Multiwfn.

## Reporting summary

Further information on research design is available in the Nature Portfolio Reporting Summary linked to this article.

## Data availability

All data supporting the findings of this study are available in this paper and the Supplementary Information. Source data are provided with this paper.

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

## Acknowledgements

This work was supported by the National Key Research and Development Program of China (2022YFF0710000 to Y. T.), the National Natural Science Foundation of China (22004037 to Z. C. L., 21811540027 and 22393930 to Y. T.), the Innovation Program of Shanghai Municipal Education Commission (201701070005E00020 to Y. T.), and 2022 Shanghai "Science and Technology Innovation Action Plan" Fundamental Research Project (22JC1401200 to Y. T.) and Fundamental Research Funds for the Central Universities (to Y. T.). The authors also thank the Materials Characterization Center of East China Normal University for helping with cell imaging.

## Author contributions

S.Z., Z.C.L. and Y. T. designed the experiments and wrote the manuscript. S.Z., Y.M., J.L. and Z.L. performed the experiments. Y.M. and J.L. contributed to synthesizing the molecules. Z.L. helped with the neuron experiments, analyzed the data of Raman detection and cellular imaging, and revised the manuscript. J.L. helped with the analysis of the data of the Raman enhancement mechanism. All the authors discussed the results and commented on the manuscript.

## Competing interests

The authors declare no competing interests.
