## [Peer Review File · Nature Communications]

Reviewers' Comments:

Reviewer #1:

Remarks to the Author:

Tian and co-workers reported the syntheses of alkyne-tagged SERS probes and their application in determining Cu^+ and Cu^{2+} ions with high sensitivity and accuracy. With the help of theoretical calculation, the authors prepared a series of ligands containing different diacetylene derivatives which provide distinguishable Raman peaks in the cellular silent region. These ligands were then used to modify the surface of gold nanostars, together with EETP as internal reference and TPP for targeting mitochondria. They found that such single probe can simultaneously detect Cu^+ and Cu^{2+} in mitochondrial lysates and in mitochondria. By qualifying the mitochondrial Cu^+ and Cu^{2+} concentrations upon ischemia, the authors proposed two pathways to explain the increase of Cu^+ and Cu^{2+} ions during the neuron death process. The idea of designing single SERS probes for dual detections of both Cu^+ and Cu^{2+} is novel and this work could be interesting to various fields, ranging from chemistry, materials science, to molecular biology and neurology. The manuscript might be acceptable for publishing after addressing the comments below.

- 1) The authors selected Cu1R5 and Cu2R1 for specific recognition of Cu^+ and Cu^{2+} . However, there is no data and discussion on the binding constants of the copper ions to the probes.
- 2) In Figure 1b, simulated Raman spectra were given. How about the experimental data? It would be more convincing if experimental data are presented.
- 3) For Raman study with neuron cells, the intensity ratio of two peaks were presented in the manuscript (such as Figure 3e, 4b, and 5d). As shown in Figure 3d, the SERS ratios varied in the cells. The authors need to specify how the data in Figures 3e, 4b, and 5d were determined.
- 4) The authors stated that "To understand the reason for ischemia-induced increase in mitochondrial copper ions, two possible pathways for the changes in the concentrations of copper ions were further investigated, including the copper ions released from copper-containing proteins as well as the conversion between Cu^+ and Cu^{2+} ." It is worth noting that conversion between Cu^+ and Cu^{2+} will NOT cause overall increase in copper ions in mitochondria.
- 5) In Figure 4g, the authors showed the release of copper ions as copper-containing proteins are destroyed. Is there any information on the initial $\text{Cu}^+/\text{Cu}^{2+}$ ratio of such protein damage? How does that ratio compare to the experimental data?
- 6) The author suggested that Cu^+ was converted into Cu^{2+} in neuronal death caused by copper overload. However, such claim cannot be fully justified by their data. As they mentioned on page 15, both Cu^+ and Cu^{2+} concentrations were increased. It is unsure how that would indicate that Cu^+ converts into Cu^{2+} .
- 7) The authors said that conversion of Cu^+ to Cu^{2+} was accompanied by the generation of ROS. If so, removing ROS in cells should promote the Cu^+ to Cu^{2+} conversion, right? This referee is confused on how the adding of GSH (ROS scavenger) can prevent the Cu^+ to Cu^{2+} conversion.
- 8) The authors found that the ROS in mitochondria was increased. Is it possible that the increase of ROS is caused by some processes during the neuronal death other than the Cu^+ to Cu^{2+} conversion?

Reviewer #2:

Remarks to the Author:

This study reports the description of an innovative system based on gold nanoparticles (gold nanostars) capable of simultaneously detecting the concentration of copper ions in both the +1 and +2 oxidation states at the mitochondrial level. The gold particles have been appropriately functionalized to carry systems that could selectively bind Cu^{2+} or Cu^+ and provide a Raman signal in a region not crowded by other signals that could arise from biological compounds. The

implications of such a designed probe and, consequently, the selective quantification of copper ions can be broad and significant, paving the way for understanding various mechanisms involved in multiple pathologies and in the mechanism of cell death known as cuproptosis, recently reported and generating considerable interest on multiple fronts. In my opinion, determining the copper ions Cu^{2+} and Cu^+ could provide important insights into cuproptosis, evidently linked to tumor proliferation and metastasis. Therefore, I believe that this work could have wide-ranging implications for research in this field and appeal to a diverse readership. The data presented in the results strongly support the claims made in the article. The data appear highly reasonable in light of the study design and conducted experiments. The description of the data in the text precisely corresponds to the figures. The quality of the figures is suitable for publication, and they can be comprehended as standalone display elements. The data are presented in figures and tables with clarity and accuracy. The figures contain high-quality images. All measurements have been repeated, and the statistics are consistent. However, I think some aspects need improvement before publication in a prestigious journal like Nature Communications.

- 1) Gold nanostars, both functionalized and non-functionalized, should be more extensively characterized under physiological conditions, as these are the conditions in which they have been applied. It is therefore recommended to determine the hydrodynamic radius of the fully functionalized system, the non-functionalized system, and various functionalization intermediates. In addition to the hydrodynamic radius, which is more relevant given the applications compared to sizes determined by TEM, it is advisable to also determine the polydispersity index to gain insights into the homogeneity of the populations.
- 2) The z-potential should be calculated to obtain information about the stability of the system at different pH levels, including physiological pH 7.4 and the pH of the mitochondrial matrix, which should be around 7.8.
- 3) The authors provide a rationale for not comparing with gold nanostars functionalized solely with $\text{Cu}1\text{R}5$ or $\text{Cu}2\text{R}1$. Interesting comparisons could have been made to understand how the detection limit could be further lowered, also in the complete system.
- 4) The authors investigate copper complexes with $\text{Cu}1\text{R}5$ and $\text{Cu}2\text{R}1$ through mass spectrometry. However, despite MALDI being a soft technique, mass spectrometry may not accurately represent the species actually present in solution. The formation of the ML complex seems plausible for the $\text{Cu}1\text{R}5$ ligand, but the 1:1 species formation is not as straightforward in the case of $\text{Cu}2\text{R}1$. Perhaps UV-vis spectroscopy, employing the method of molar ratio, could be more indicative (clearly at physiological pH or in a mixed solvent if the aqueous solubility of the ligand or the complex/ligand is not optimal)
- 5) Analyzing how the hydrodynamic radius varies during titration with copper(I)/copper(II) could provide insights into various particle aggregation phenomena. These aggregations might occur at the mitochondrial level, subsequently influencing the signal detected for metal quantification.

Supplementary Material

- In the paragraph titled "Chemicals and Reagents," it is not clear which Cu^{2+} salt was used for titrations to assess the probe's responsiveness to Cu^{2+} concentration. Another important piece of information could pertain to the Cu^{2+} solution: whether it was purchased as is at a known concentration (potentially diluted), or if the exact concentration of the solution was determined after preparation by weighing a salt.

- Supplementary Fig. 1

The authors should review and correct the caption for Figure 1, panel b.

"The synthesis procedures of molecules. a, The synthesis procedures of $\text{Cu}1\text{R}5$. b, The synthesis procedures of $\text{Cu}1\text{R}5$."

- Supplementary Fig 3-5-6-8-9-12-16 It is recommended to zoom in or cut the x-axis to clearly show and attribute the peaks.

- Supplementary Fig 27

The concentration of ES should be indicated in the caption for both panel a and panel b. In panel b, it is also necessary to clarify the copper concentration so that the reader has a clear idea upon initial inspection of the figure

Minor revisions

In the section " Methods": Line 736. Check the meaning, What is AA?

Overall, the manuscript could be accepted for publication in this journal after these revisions are taken into account.

Reviewer #3:

Remarks to the Author:

This manuscript developed two Raman reporters which can respond Cu^+ and Cu^{2+} , respectively. The authors used the mitochondrial-specific plasmonic nanoprobe for many scenes, such as, oxidative stress, and drug treatment, and show time-dependent changes. This manuscript is well organized and the data support the conclusion. Before it is acceptable, several issues should be addressed.

1. In the introduction part, the author stated that "SERS probes assembled based on Au-S bonds are easily susceptible to unpredictable aggregation in cells". To support this, some references should be added.
2. Line 301-305, the sentence "Previous literatures has reported that intracellular copper ions were almost bound and stored by various metal-binding proteins rather than free copper ions, to maintain copper homeostasis and avoid free copper ions-involved oxidative stress damage in cells. 38, 39 However, we found a remarkable accumulation of mitochondrial copper ions during ischemia." is wired. The point of the literature is to note that the storage state is metal-binding protein. I didn't get why the author emphasised the accumulation in mitochondria. Is that mean that no metal-binding protein in mitochondria? Please rewrite it here.
3. The characterization of the SERS probe, some details are missing, e.g., Zeta potential, UV-vis spectra, TEM image, etc.
4. DFT calculation should be supplemented for Cu^+ and Cu^{2+} , which are also needed for supporting the band assignments of their complex in response to Cu^+ and Cu^{2+} (Supplementary Table. 1).
5. More experimental details for SERS imaging should be added, such as the accumulation time, objective lens, etc. Did you use Raman mapping mode or imaging mode to obtain the imaging data? If the imaging mode you use, please show the filter information of each channel.
6. In Figure 3a, it seems CuPM@GN shows strong fluorescence (green color). Did this effect impact SERS measurement?
7. The bands 2213 and 2238 cm^{-1} seem overlaid to a certain extent. I suggest using the peak fit function to show their intensities. And Figure 2d-f should be re-plotted according to the fitted data.

Reply to Reviewer 1

Tian and co-workers reported the syntheses of alkyne-tagged SERS probes and their application in determining Cu^+ and Cu^{2+} ions with high sensitivity and accuracy. With the help of theoretical calculation, the authors prepared a series of ligands containing different diacetylene derivatives which provide distinguishable Raman peaks in the cellular silent region. These ligands were then used to modify the surface of gold nanostars, together with EETP as internal reference and TPP for targeting mitochondria. They found that such single probe can simultaneously detect Cu^+ and Cu^{2+} in mitochondrial lysates and in mitochondria. By qualifying the mitochondrial Cu^+ and Cu^{2+} concentrations upon ischemia, the authors proposed that two pathways to explain the increase of Cu^+ and Cu^{2+} ions during neuron death process ischemia. The idea of designing single SERS probes for dual detections of both Cu^+ and Cu^{2+} is novel and this work could be interesting to various fields, ranging from chemistry, materials science, to molecular biology and neurology. The manuscript might be acceptable for publishing after addressing the comments below.

Q1-1. The authors selected Cu^1R_5 and Cu^2R_1 for specific recognition of Cu^+ and Cu^{2+} . However, there is no data and discussion on the binding constants of the copper ions to the probes.

A1-1. We greatly appreciate the reviewer for high evaluation and valuable suggestions. The binding constants of the copper ions to the probe were further obtained. The stoichiometry composition of $\text{Cu}^1\text{R}_5\text{-Cu}^+$ complex formed between Cu^+ ion and Cu^1R_5 ligand was first determined by absorbance spectra measurements. As shown in Figure R1-1a, Cu^+ aqueous solution with different concentrations was added into a cuvette containing 20.0 μM of Cu^1R_5 ligand dissolved in an acetonitrile/phosphate ($\text{CH}_3\text{CN}/\text{PBS}$) buffer (1:1, v/v; 20 mM PBS buffer, pH = 7.40) mixture. A new band around 386 nm was observed after addition of Cu^+ , which was attributed to the extension of the electron conjugation system by the chelate structure formed after the coordination of the metal ion with the ligand, resulting in the emergence of new absorption peaks at longer wavelengths (*Inorg. Chem.* 2011, 50, 1213-1219). In addition, an iso-absorption point was observed at 330 nm, which appeared between the peak maximum of Cu^1R_5 (312 nm) and the peak maximum of the complex (386 nm), suggesting a single equilibrium step (*J. Fluoresc.* 2023, 33, 1003-1015). More importantly, with the concentration of Cu^+ increased from 0 to 20.0 μM , the intensity of the absorption band around 386 nm was gradually increased. No obvious spectrophotometric change was observed after the concentration of

Cu⁺ was higher than 20.0 μM. As summarized in Figure R1-1b, the stoichiometry ratio between Cu⁺ ion and Cu¹R₅ ligand was determined as 1:1. Then, the binding constant K of Cu¹R₅-Cu⁺ complex was determined using Benesi-Hildebrand equation (*Org. Lett.* 2008, 10, 1481–1484) as shown in Equation (1) (Eq. 1),

$$\frac{1}{A-A_0} = \frac{1}{A_{\max}-A_0} + \frac{1}{K[M]} \times \frac{1}{A_{\max}-A_0} \quad (1)$$

where K was the binding constant, A₀ and A represented the observed absorption in the absence and in the presence of ions, respectively. A_{max} was the observed absorption when ions reached saturation and [M] was the concentration of added ions. As shown in Figures R1-1c, 1/(A-A₀) and 1/[Cu⁺] had a good linear relationship. Thus, the binding constant of Cu¹R₅ with Cu⁺ was calculated as (5.65±0.57) ×10⁴ (n= 5, S. D.)

On the other hand, the stoichiometry ratio between Cu²⁺ ion and Cu²R₁ ligand was also obtained by a similar method. As shown in Figure R1-1d, a new band around 404 nm was observed upon the addition of Cu²⁺, and an iso-absorption point was observed at 362 nm, which appeared between the peak maximum for Cu²R₁ (~306 nm) and the peak maximum of the complex (410 nm), suggesting that it was formed by a single equilibrium step. The stoichiometry ratio between Cu²⁺ ion and Cu²R₁ ligand was determined as 1:1 (Figure R1-1e). Moreover, the binding constant of Cu²R₁ with Cu²⁺ was calculated as (5.61±0.94) ×10⁴ (n= 5, S. D.). The discussion was added in the revised manuscript (Page 7, Lines 26-30 and Page 8 Lines 1-15); Figure R1-1 was added in the revised Supporting Information (Page S22, Supplementary Fig. 30).

Figure R1-1. (a) Absorption spectra of Cu¹R₅ (20 μM) upon titration with [Cu(CH₃CN)₄][PF₆] from 0 μM (0 eq.) to 30 μM (1.5 eq.) (0, 1.5, 2.0, 4.0, 6.0, 8.0, 10.0, 12.0, 16.0, 20.0, 24.0, 25.0 and 30.0 μM) in a CH₃CN/PBS buffer (1:1, v/v; 20 mM PBS buffer, pH = 7.40) mixture. The peak associated with formation of the complex is at λ_{max} = 386 nm. (b) A plot of absorbance at 386 nm as a function of mole ratio (Cu⁺: Cu¹R₅) where the tangents intersect at the metal: complex ratio (Error bars: S. D., n=5). (c) Benesi–Hilderbrand plot of Cu¹R₅ with addition of Cu⁺ (2.0-20.0 μM, Error bars: S. D., n=5). (d) Absorption spectra of Cu²R₁ (20 μM) upon titration with CuCl₂·2H₂O from 0 μM (0 eq.) to 30 μM (1.5 eq.) (0, 1.5, 2.0, 4.0, 6.0, 8.0, 10.0, 12.0, 16.0, 20.0, 24.0, 25.0 and 30.0 μM) in a CH₃CN/PBS buffer (1:1, v/v; 20 mM PBS buffer, pH = 7.40) mixture. The peak associated with formation of the complex is at λ_{max} = 404 nm. (e) A plot of absorbance at 410 nm as a function of mole ratio (Cu²⁺: Cu²R₁) where the tangents intersect at the metal: complex ratio (Error bars: S. D., n=5). (f) Benesi–Hilderbrand plot of Cu²R₁ with Cu²⁺ (2.0-20.0 μM, Error bars: S. D., n=5).

Q1-2. In Figure 1b, simulated Raman spectra were given. How about the experimental data? It would be more convincing if experimental data are presented.

A1-2. For simultaneous recognition of Cu⁺ and Cu²⁺, a series of Raman molecules for specific recognition of Cu⁺ or Cu²⁺ were designed, which contained diacetylene derivatives (R₁-R₅) as Raman reporters at cellular Raman-silent region. According to classical mechanics model shown in Eq.2 (*J. Am. Chem. Soc.* 2014, 136, 8027–8033),

$$v = \frac{1}{2\pi c} \sqrt{\frac{k}{\mu}}, \text{ where } \mu = \frac{m_1 m_2}{m_1 + m_2} \quad (2)$$

where v denotes the wavenumber of the stretching vibration belonging to a diacetylene, m_1 and m_2 represent the atomic masses at both ends of the diacetylene bond, and k denotes the force constant of the diacetylene bond. It was considered that π -electron on diacetylene may be influenced after the alkyls (R₁) were replaced by benzyl and benzamide simultaneously (R₅), and the stretching force constant of the vibrational mode may be weakened. Thus, Raman peak attributed to diacetylene stretching vibration may be significantly redshifted.

According to Raman spectra of Cu¹R_x and Cu²R_x obtained by theoretical simulation shown in Figure 1b in our original manuscript, Cu¹R₅ and Cu²R₁, with clearly distinguishable Raman signals in cellular silent region (2213 cm⁻¹ belonging to Cu¹R₅ and 2238 cm⁻¹ belonging to Cu²R₁), were selected for synthesis for specific recognition of Cu⁺ and Cu²⁺, respectively. Then, Cu¹R₅ and Cu²R₁ were synthesized, and Raman spectra of Cu¹R₅ and Cu²R₁ were

experimentally obtained by conjugating them onto gold nanostars (GNs). As shown in Figure 1d in our original manuscript, Raman peaks of Cu¹R₅ and Cu²R₁ were measured at 2213 cm⁻¹ and 2238 cm⁻¹, respectively, consistent well with those of theoretical calculation results.

Furthermore, according to the suggestion of the reviewer, Cu¹R₃ and Cu²R₃ were further synthesized to verify the reliability of theoretical calculation results (Figure R1-2), because the calculated Raman peaks of Cu¹R₃ and Cu²R₃ were located in the middle positions between Cu¹R₅ and Cu²R₁. The structures of Cu¹R₃ and Cu²R₃ were confirmed by ¹H NMR spectra and mass spectra, respectively (from Figure R1-3 to Figure R1-6). Then, Raman spectra of Cu¹R₃ and Cu²R₃ were also experimentally obtained by conjugating them onto gold nanostars (GNs), respectively. As shown in Figure R1-7, Raman peaks of Cu¹R₃ and Cu²R₃ were obtained at 2222 cm⁻¹ and 2226 cm⁻¹, respectively, consistent with those of theoretical calculation results shown in Figure 1b in our original manuscript. The experimental results of the Raman peaks of these typical molecules including Cu¹R₅, Cu²R₁, Cu¹R₃ and Cu²R₃ were in agreement with those obtained by theoretical calculation results. Therefore, the simulated Raman spectra of these molecules were convincing. The supplementary results were added in the revised manuscript (Page 5, Lines 6-11; Page 28, Lines 25-30; Page 29, Lines 1-29, and Page 30, Lines 1-6) and the revised Supporting Information (Page S16, Supplementary Fig. 20; Page S17, Supplementary Figs. 21-22 and Page S18, Supplementary Figs.23-24).

Figure R1-2. The synthesis procedures of Cu¹R₃ and Cu²R₃.

Figure R1-3. ^1H NMR of Cu^1R_3 . ^1H NMR (600 MHz, CDCl_3) δ (ppm): 8.01-8.00 (d, 2H), 7.59-7.57 (dd, 2H), 6.50-6.48 (t, 1H), 4.23-4.22 (dd, 2H), 3.65 (s, 2H), 2.74-2.64 (m, 16H), 2.57-2.53 (q, 4H), 2.26-2.25 (t, 1H), 1.27-1.24 (t, 6H).

Figure R1-4. Mass spectrum of Cu^1R_3 . HR-MS for $\text{C}_{27}\text{H}_{36}\text{N}_2\text{NaOS}_4$ was found to be 555.1927.

Figure R1-5. ^1H NMR of Cu^2R_3 . ^1H NMR (600 MHz, CDCl_3) δ (ppm): 8.57-8.56 (dt, 2H), 8.05-8.04 (d, 2H), 7.67-7.65 (td, 2H), 7.53-7.52 (d, 2H), 7.39-7.36 (dd, 2H), 7.18-7.15 (m, 2H), 6.51-6.48 (t, 3H), 4.83-4.81 (t, 2H), 3.93 (s, 4H), 3.44 (s, 2H), 2.31-2.30 (t, 1H).

Figure R1-6. Mass spectrum of Cu^2R_3 . HR-MS for $\text{C}_{27}\text{H}_{22}\text{N}_4\text{NaO}$ was found to be 441.1835.

Figure R1-7. (a) SERS spectra of $\text{Cu}^1\text{R}_3@GN$ (I) and theoretically simulated Raman spectra of Cu^1R_3 (II). (b) SERS spectra of $\text{Cu}^2\text{R}_3@GN$ (I) and theoretically simulated Raman spectra of Cu^1R_3 (II).

Q1-3. For Raman study with neuron cells, the intensity ratio of two peaks were presented in the manuscript (such as Figure 3e, 4b, and 5d). As shows in Figure 3d, the SERS ratios varied in the cells. The authors need to specify how the data in Figures 3e, 4b, and 5d were determined.

A1-3. Figure 3d in our original manuscript was obtained by Raman mapping after the probe entered into the cell. The SERS ratios (I_{2213}/I_{2155} and I_{2238}/I_{2155}) varied in cells in Figure 3d, indicating that concentrations of mitochondrial Cu^+ and Cu^{2+} were different in cells. The data shown in Figures 3e, 4b and 5d in our original manuscript were the mean values of SERS ratios (I_{2213}/I_{2155} and I_{2238}/I_{2155}) in mitochondria under different stimulations, SERS ratios (I_{2213}/I_{2155} and I_{2238}/I_{2155}) were statistical results obtained from more than 10 neurons (*Anal. Chem.* 2016, 88, 9518-9523). Specifically, Figure 3e shows the statistical results of I_{2213}/I_{2155} and I_{2238}/I_{2155} ratios in mitochondria under the stimulation of exogenous copper ions. Figure 4b shows the statistical results of I_{2213}/I_{2155} and I_{2238}/I_{2155} ratios in mitochondria under the stimulation of hypoxia and sugar deficiency (OGD). Figure 5d shows the statistical results of I_{2213}/I_{2155} and I_{2238}/I_{2155} ratios in mitochondria under the stimulation of copper overload. The results showed that the levels and distribution of mitochondrial Cu^+ and Cu^{2+} in neurons were different under different stimulations. The explanation was added in the revised manuscript (Page 11, Lines 22-24).

Q1-4. The authors stated that “To understand the reason for ischemia-induced increase in mitochondrial copper ions, two possible pathways for the changes in the concentrations of copper ions were further investigated, including the copper ions released from copper-containing proteins as well as the conversion between Cu^+ and Cu^{2+} .” It is worth noting that conversion between Cu^+ and Cu^{2+} will NOT cause overall increase in copper ions in mitochondria.

A1-4. We greatly appreciate the reviewer’s suggestion. According to our experimental results, both of free Cu^+ and Cu^{2+} in ischemic neurons stimulated by hypoxia and sugar deficiency (OGD) were significantly increased (Cu^+ increased to $\sim 3.09 \mu\text{M}$ and Cu^{2+} increased to $\sim 6.44 \mu\text{M}$), and the main reasons for this phenomenon were the structural destruction of copper-containing proteins and the release of bound copper. It should be pointed out that the concentration of Cu^{2+} increased ($\sim 6.44 \mu\text{M}$) more than that of Cu^+ ($\sim 3.09 \mu\text{M}$) during OGD treatment. Interestingly, as shown in Figure R1-8, the level of free Cu^{2+} only increased to $\sim 3.77 \mu\text{M}$, after the neurons were treated with OGD in the presence of bathocuproine disulfonate (BCS), a Cu^+ chelator, while seldom increase was observed for Cu^+ ($< 0.32 \pm 0.03 \mu\text{M}$, which was the LOD for Cu^+).

These results strongly proved that the increased Cu^{2+} partly came from Cu^+ conversion during ischemia.

Overall, the conversion between Cu^+ and Cu^{2+} could not lead to an increase in the overall level of copper ions in mitochondria, but it was the main reason for the increase in intracellular Cu^{2+} during ischemia. As pointed out by the reviewer, the sentence “To understand the reason for ischemia-induced increase in mitochondrial copper ions” was corrected to “To understand the reason for the increase in Cu^{2+} concentration higher than that of Cu^+ during ischemia” in the revised manuscript (Page 14, Lines 12-16). Figure R1-8 was added in the revised manuscript (Page 15, Fig. 4).

Figure R1-8. (a) Bright-field images of neurons (left) and SERS imaging of Cu^+ channel (I_{2213}/I_{2155} , middle) and Cu^{2+} channel (I_{2238}/I_{2155} , right) in neurons pre-treated with BCS (20 μM) for 12 h and then treated with OGD for 3 h. (b) SERS intensity ratios of I_{2213}/I_{2155} and I_{2238}/I_{2155} were obtained from Figure R1-8 a ($n=10$, S. D.).

Q1-5. In Figure 4g, the authors showed the released of copper ions as the copper-containing proteins are destroyed. Is there any information on the initial $\text{Cu}^+/\text{Cu}^{2+}$ ratio of such protein damage? How does that ratio compare to the experimental data?

A1-5. Copper is too redox-active to exist in an unbound form in the cell without causing oxidative damage (*Nature*, 2010, 465, 645-648). Maintenance of cellular copper homeostasis requires membrane copper transporters and a family of proteins, termed “copper chaperones” that deliver copper to specific targets in the cell (*Chem. Rev.* 2006, 106, 1995-2044). According to previous literature reports, an upper limit of 10^{-18} M for the free concentration of copper ions in unstressed cells has been evaluated (*Science* 1999, 284, 805). Taking into account the volume of a single cell ($\sim 10^{-14}$ L), it was deduced that cellular

free copper ions were negligible which was widely recognized (*Science* 1999, 284, 748).

By using our developed CuPM@GN probe, as shown in Figure 4b in our original manuscript, almost no copper ions were detected in the mitochondria of normal neurons, indicating the levels of intracellular free Cu^+ and Cu^{2+} were much lower than the detection limit (LOD) of our developed Raman probe (LOD for Cu^+ and Cu^{2+} were $0.32\pm 0.03 \mu\text{M}$ and $0.44\pm 0.02 \mu\text{M}$, respectively). Therefore, our experimental results are consistent with the previous literature reports.

Q1-6. The author suggested that Cu^+ was converted into Cu^{2+} in neuronal death caused by copper overload. However, such claim cannot be fully justified by their data. As they mentioned on page 15, both Cu^+ and Cu^{2+} concentrations were increased. It is unsure how that would indicate that Cu^+ converts into Cu^{2+} .

A1-6. As shown in Fig. 5c in our original manuscript, both the concentrations of mitochondrial Cu^+ and Cu^{2+} were gradually increased after the neurons were stimulated by CuCl_2 in the presence of elesclomol (ES), a copper ionophore (ES-Cu). The concentration of Cu^+ (C_{Cu^+}) in mitochondria increased from a very low level ($< 0.32\pm 0.03 \mu\text{M}$, which was the LOD for Cu^+) to $\sim 2.35\pm 0.43 \mu\text{M}$, while that of mitochondrial Cu^{2+} ($C_{\text{Cu}^{2+}}$) was increased from a very low level ($< 0.44\pm 0.02 \mu\text{M}$, which was the LOD for Cu^{2+}) to $\sim 5.89\pm 1.04 \mu\text{M}$. Notably, the concentration of Cu^{2+} increased ($\sim 5.89 \mu\text{M}$) more than that of Cu^+ ($\sim 2.35 \mu\text{M}$) during the stimulation of extracellular ES-Cu complex. Interestingly, as shown in Figure R1-9a, the level of free Cu^{2+} only increased to $\sim 1.06 \mu\text{M}$, after the neurons were treated with ES-Cu complex in the presence of BCS, while seldom increase was observed for Cu^+ ($< 0.32\pm 0.03 \mu\text{M}$, which was the LOD for Cu^+) (Figure R1-9b). These results strongly proved that the increased Cu^{2+} partly came from Cu^+ conversion during the process.

Moreover, it was found that the concentration of superoxide anion ($\text{O}_2^{\cdot-}$) in mitochondria also gradually increased from $2.02\pm 0.28 \mu\text{M}$ to $14.75\pm 1.30 \mu\text{M}$ during the process. Interestingly, it was observed that the concentration of $\text{O}_2^{\cdot-}$ in mitochondria was decreased to $3.97\pm 0.71 \mu\text{M}$ after the neurons were treated with ES-Cu complex in the presence of BCS (Figure R1-9c). The results further confirmed that the conversion of Cu^+ to Cu^{2+} occurred during copper overload, which was accompanied by $\text{O}_2^{\cdot-}$ burst simultaneously.

To further prove that Cu^+ converts into Cu^{2+} under this model, different concentrations of GSH (0.1 mM and 1.0 mM) were used to scavenge ROS in cells. As shown in Fig. 5d in our original manuscript, after the neurons were stimulated by overload copper ions for 24 h in the presence of a low

concentration of GSH (0.1 mM), C_{Cu2} reduced by $\sim 17.20\%$ (to $4.90 \pm 0.48 \mu M$), while C_{Cu1} increased by $\sim 15.13\%$ (to $3.24 \pm 0.37 \mu M$), compared with those of the copper overload neurons without GSH. Moreover, with the concentration of the stimulated GSH increased to 1.0 mM, C_{Cu2} further reduced to 3.95 ± 0.39 ($\sim 33.26\%$), while C_{Cu1} increased to $3.44 \pm 0.44 \mu M$ ($\sim 22.14\%$). More interestingly, as shown in Fig. 5e in our original manuscript, the concentration of $O_2^{\cdot -}$ in mitochondria was decreased by $\sim 38.48\%$ (to $9.85 \pm 0.87 \mu M$) and $\sim 74.64\%$ (to $4.06 \pm 0.71 \mu M$), after the neurons were stimulated by overload copper ions for 24 h in the presence of 0.1 mM and 1.0 mM GSH, respectively. These results strongly proved that Cu^+ was converted into Cu^{2+} in neuronal death caused by copper overload. The corresponding discussion was added in the revised manuscript (Page 17, Lines 12-16 and Lines 23-30); Figure R1-9 was added in the revised manuscript (Page 20, Fig. 5) and the revised Supporting Information (Page S27, Supplementary Fig. 38).

Figure R1-9. (a) Bright-field images of neurons (left) and SERS imaging of Cu^+ channel (I₂₂₁₃/I₂₁₅₅, middle) and Cu^{2+} channel (I₂₂₃₈/I₂₁₅₅, right) in neurons during 100 nM elesclomol-CuCl₂ (ES-Cu, 1:1)-induced copper overload. For comparison, neurons were treated with ES-Cu in the presence of 20 μM BCS. (b) The SERS intensity ratios of I₂₂₁₃/I₂₁₅₅ and I₂₂₃₈/I₂₁₅₅ were obtained from different stimulations in Figure R1-9a (n=10, S. D.). (c) The concentrations of $O_2^{\cdot -}$ in mitochondria were obtained from different stimulations in Figure R1-9a (n=10, S. D.).

Q1-7. The authors said that conversion of Cu^+ to Cu^{2+} was accompanied by the generation of ROS. If so, removing ROS in cells should promote the Cu^+ to Cu^{2+} conversion, right? This referee is confused on how the adding of GSH (ROS scavenge) can prevent the Cu^+ to Cu^{2+} conversion.

A1-7. It was necessary to confirm that the addition of GSH prevented the oxidation of Cu^+ to generate Cu^{2+} . As we answered in A1-6, the results strongly proved that the conversion of Cu^+ to Cu^{2+} occurred during copper overload, while O_2 was reduced to $O_2^{\cdot -}$ and other ROS, simultaneously. Firstly, it should

be noted that the conversion of Cu^+ to Cu^{2+} and the generation of reactive oxygen species (ROS) were two processes that occurred simultaneously. The conversion of Cu^+ to Cu^{2+} was initiated by oxidants, which include oxygen (O_2) and other oxidizing molecules (*J. Hazard. Mater.* 2018, 344, 1209-1219). In this process, Cu^+ ions transitioned to Cu^{2+} by losing an electron, and the oxidizing molecules transformed into ROS by gaining an electron, such as O_2 converted to superoxide anions ($\text{O}_2^{\cdot-}$) (*J. Hazard. Mater.* 2014, 275, 193-199). The two processes were independently separate but mutually influenced each other, especially since they could not be entirely understood as one process.

On the other hand, as an antioxidant, glutathione (GSH) primarily exerted its strong reducing effects and eliminated ROS through the process of donating electrons from its thiol group (-SH) (*Prog. Neurobiol.* 2000, 62, 649-671). Highly active and oxidizing reactive oxygen molecules, such as $\text{O}_2^{\cdot-}$, were transformed into less-oxidative molecules like water under the influence of GSH. For another, GSH could substitute for Cu^+ to provide electrons to oxidant molecules and help counteract oxidative effects in the oxidative environment (*Water Sci. Technol.* 2018, 78, 1390-1399; *RSC Adv.* 2016, 6, 38541-38547). Above all, addition of GSH could prevent the conversion of Cu^+ to Cu^{2+} , which was in a good agreement with our experimental results.

Q1-8. The authors found that the ROS in mitochondria was increased. Is that possible that the increase of ROS is caused by some processes during the neuronal death other than the Cu^+ to Cu^{2+} conversion?

A1-8. As we mentioned in A1-6, the results proved that the conversion of Cu^+ to Cu^{2+} occurred during copper overload, while O_2 was reduced to $\text{O}_2^{\cdot-}$ and other ROS, simultaneously. It should be pointed out that the concentration of $\text{O}_2^{\cdot-}$ in mitochondria was decreased to $3.97 \pm 0.71 \mu\text{M}$ after the neurons were treated with ES-Cu complex in the presence of BCS (Figure R1-9c), which was still higher than that in neurons without 100 nM ES-Cu-treated ($2.02 \pm 0.28 \mu\text{M}$). The results suggested that the process of Cu^+ to Cu^{2+} conversion, as well as other processes during neuronal death may also contribute to the increase of mitochondrial $\text{O}_2^{\cdot-}$, such as the leakiness of the electron transport chain (ETC) and the abnormal synthesis of ATP (*Nat. Rev. Microbiol.* 2017, 15, 385-396; *Cell* 2008, 134, 279-290). The corresponding discussion was added in the revised manuscript (Page 17, Line 30, and Page 18, Lines 1-3)

Reply to Reviewer 2

This study reports the description of an innovative system based on gold nanoparticles (gold nanostars) capable of simultaneously detecting the concentration of copper ions in both the +1 and +2 oxidation states at the mitochondrial level. The gold particles have been appropriately functionalized to carry systems that could selectively bind Cu^{2+} or Cu^+ and provide a Raman signal in a region not crowded by other signals that could arise from biological compounds. The implications of such a designed probe and, consequently, the selective quantification of copper ions can be broad and significant, paving the way for understanding various mechanisms involved in multiple pathologies and in the mechanism of cell death known as cuproptosis, recently reported and generating considerable interest on multiple fronts. In my opinion, determining the copper ions Cu^{2+} and Cu^+ could provide important insights into cuproplasia, evidently linked to tumor proliferation and metastasis. Therefore, I believe that this work could have wide-ranging implications for research in this field and appeal to a diverse readership. The data presented in the results strongly support the claims made in the article. The data appear highly reasonable in light of the study design and conducted experiments. The description of the data in the text precisely corresponds to the figures. The quality of the figures is suitable for publication, and they can be comprehended as standalone display elements. The data are presented in figures and tables with clarity and accuracy. The figures contain high-quality images. All measurements have been repeated, and the statistics are consistent. However, I think some aspects need improvement before publication in a prestigious journal like Nature Communications.

Q2-1. Gold nanostars, both functionalized and non-functionalized, should be more extensively characterized under physiological conditions, as these are the conditions in which they have been applied. It is therefore recommended to determine the hydrodynamic radius of the fully functionalized system, the non-functionalized system, and various functionalization intermediates. In addition to the hydrodynamic radius, which is more relevant given the applications compared to sizes determined by TEM, it is advisable to also determine the polydispersity index to gain insights into the homogeneity of the populations.

A2-1. We are grateful for the reviewer's positive evaluation. As suggested by the reviewer, we supplemented the dynamic light scattering (DLS) of the fully functionalized gold nanostars (GNs), the non-functionalized GNs and various functionalization intermediates in PBS (20 mM, pH 7.4), using a Malvern Zetasizer (Nano ZS3600, Malvern, UK). As shown in Figure R2-1, non-

functionalized GNs showed a hydrodynamic diameter of 54.47 ± 0.28 nm ($n = 5$, S. D.). In addition, hydrodynamic diameters of various functionalization intermediates including [4-oxo-4-(2-propyn-1-ylamino) butyl] triphenylphosphonium (TPP), 1-ethynyl-4-[2-[tris (1-methylethyl) silyl] ethynyl] benzene (EETP), Cu^1R_5 or Cu^2R_1 functionalized GNs (TPP@GN, EETP@GN, Cu^1R_5 @GN or Cu^2R_1 @GN) were also obtained, which were calculated as 65.23 ± 1.61 nm, 60.49 ± 0.39 nm, 80.66 ± 3.71 nm and 64.88 ± 0.55 nm, respectively. Interestingly, after Cu^1R_5 , Cu^2R_1 , TPP and ETPP were simultaneously modified on the surface of GNs to form CuPM@GN probe, the hydrodynamic diameter of CuPM@GN probe was further increased to 87.30 ± 1.69 nm ($n = 5$, S. D.).

On the other hand, as shown in Figure R2-1, polydispersity index (PDI) of GNs was obtained as 0.211, while those of various functionalization intermediates were obtained as 0.282-0.398. Moreover, PDI of CuPM@GN probe was estimated as 0.386. Since PDI less than 0.4 suggests a low tendency for agglomeration of nanoparticles (*Int. J. Mol. Sci.* 2021, 22, 5072) the results proved that our developed probes had no obvious aggregation. The corresponding discussion was added in the revised manuscript (Page 5, Lines 27-28); Figure R2-1 was added in the revised Supporting Information (Page S20, Supplementary Fig. 26).

Figure R2-1. Size distribution and PDI characterization of (I) GNs, (II) TPP@GN, (III) EETP@GN, (IV) Cu^1R_5 @GN, (V) Cu^2R_1 @GN and (VI) CuPM@GN, respectively.

Q2-2. The z-potential should be calculated to obtain information about the stability of the system at different pH levels, including physiological pH 7.4 and the pH of the mitochondrial matrix, which should be around 7.8.

A2-2. According to the suggestions proposed by the reviewer, zeta potentials of GNs, various functionalization intermediates and CuPM@GN probe were first measured at physiological pH 7.4. As shown in Figure R2-2a, zeta potential of individual GNs was obtained as -36.1 ± 1.2 mV ($n = 5$, S. D.), which can be attributed to the abundance of negative charges provided by citrate salts on the surface of GNs. In addition, zeta potentials of ETPP, Cu¹R₅ or Cu²R₁ functionalized GNs (ETPP@GN, Cu¹R₅@GN, Cu²R₁@GN) were estimated as -33.8 ± 0.9 , -25.3 ± 2.8 and -30.5 ± 1.6 mV ($n=5$, S. D.), respectively. Interestingly, zeta potentials of TPP functionalized GNs (TPP@GNs) were estimated as 26.7 ± 2.3 mV ($n=5$, S. D.). The positive potential value of TPP@GN was a result of the modification with positively charged TPP on the surface of GNs. Moreover, zeta potential of the developed CuPM@GN probe was also estimated as 23.1 ± 2.8 mV ($n=5$, S. D.).

Figure R2-2. (a) Zeta potentials of (I) GNs, (II) TPP@GN, (III) EETP@GN, (IV) Cu¹R₅@GN, (V) Cu²R₁@GN and (VI) CuPM@GN in 20 mM PBS (pH 7.4). (b) Zeta potentials of CuPM@GN probe at different pH (5.0, 6.3, 7.0, 7.4, 7.8 and 8.0).

Furthermore, zeta potential of the developed CuPM@GN probe was further estimated at different pH conditions (5.0-8.0). As shown in Figure R2-2b, zeta potential of CuPM@GN probe was decreased from 27.9 mV to 20.8 mV, with the pH values increased from 5.0 to 8.0. It should be noted that particle suspensions with an absolute zeta potential greater than 20 mV are generally considered to have anti-aggregation stability, because the charge repulsion is greater than the van der Waals attraction between the particles (*Int. J. Mol. Sci.* 2021, 22, 5072). The results indicated that the developed CuPM@GN probe has good dispersion at the pH 5.0-8.0, this may be attributed to the charge shielding effect and a certain hydrophobicity provided by the ligand layer, which reduced sensitivity to changes in environmental pH. This pH range covers both

the normal physiological environment and mitochondrial matrix. The corresponding discussion was added in the revised manuscript (Page 5, Lines 27-28; Page 9, Lines 17-22); Figure R2-2 was added in the revised Supporting Information (Page S20, Supplementary Fig. 27; and Page 25, Supplementary Fig. 34a).

Q2-3. The authors provide a rationale for not comparing with gold nanostars functionalized solely with Cu¹R₅ or Cu²R₁. Interesting comparisons could have been made to understand how the detection limit could be further lowered, also in the complete system.

A2-3. According to the suggestion proposed by the reviewer, Cu¹R₅- or Cu²R₁-functionalized probe was further prepared by using a similar method to that of CuPM@GN probe. Specifically, Cu¹R₅, TPP and EETP were simultaneously conjugated onto GN with concentration ratio of 4:4:1, denoted as Cu¹R₅@GN. Meanwhile, Cu²R₁, TPP and EETP were simultaneously conjugated onto GN with concentration ratio of 4:4:1, denoted as Cu²R₁@GN.

Then, the analytical performance of Cu¹R₅@GN towards Cu⁺ and Cu²R₁@GN towards Cu²⁺ was estimated in mitochondrial lysates. As shown in Figure R2-3a, with the gradual addition of Cu⁺, the peak intensity at 2213 cm⁻¹ (*I*₂₂₁₃) of Cu¹R₅@GN was decreased, while that at 2155 cm⁻¹ (*I*₂₁₅₅) of Cu¹R₅@GN hardly changed. SERS intensity ratio between *I*₂₂₁₃ and *I*₂₁₅₅ (*I*₂₂₁₃/*I*₂₁₅₅) showed good linearity with Cu⁺ concentration in the range of 0.20-21.00 μM, and the limit of detection (LOD) was estimated to be 0.11±0.02 μM (3σ/*k*) (n=5, S. D., Figure R2-3b). Thus, LOD for Cu⁺ obtained by Cu¹R₅@GN probe was lower than that obtained by CuPM@GN probe (0.32±0.03 μM), and the detection linear range obtained by Cu¹R₅@GN probe was wider than that obtained by CuPM@GN probe (0.50-14.00 μM).

On the other hand, as demonstrated in Figure R2-3c, with increasing concentration of Cu²⁺, the peak intensity at 2238 cm⁻¹ (*I*₂₂₃₈) of Cu²R₁@GN was decreased, yet *I*₂₁₅₅ of Cu²R₁@GN remained unchanged. *I*₂₂₃₈/*I*₂₁₅₅ exhibited a good linear relationship with Cu²⁺ concentration from 0.30 to 28.00 μM, with a LOD of 0.13±0.03 μM (3σ/*k*) (n=5, S. D., Figure R2-3d). Therefore, LOD for Cu²⁺ obtained by Cu²R₁@GN probe was lower than that obtained by CuPM@GN probe (0.44±0.02 μM), and the detection linear range obtained by Cu²R₁@GN probe was wider than that obtained by CuPM@GN probe (0.50-16.00 μM).

Therefore, simultaneous detection of Cu⁺ and Cu²⁺ by using the mixed probes (Cu¹R₅@GN and Cu²R₁@GN) would result in a lower LOD than that of single CuPM@GN probe. However, as we answered in A1-5, cellular free

copper ions were negligible which was widely recognized, the analytical performance of our developed CuPM@GN probe has been able to meet the detection of intracellular Cu⁺ and Cu²⁺ simultaneously. More importantly, the use of mixed probes would require the cells to take up more nanoparticles, which was detrimental to cells. Furthermore, the mixed probes entered into cells would generate different localizations, different concentration ratios and altered metabolisms, making the scenario very complicated.

Figure R2-3. (a) SERS spectra of Cu¹R₅@GN in mitochondrial lysates with the addition of Cu⁺ with various concentrations (0.1, 0.2, 0.5, 1.0, 2.0, 5.0, 10.0, 12.0, 15.0, 18.0, 21.0 and 23.0 μM). (b) Plots of SERS intensity ratios (I₂₂₁₃/I₂₁₅₅) versus the concentration of Cu⁺ obtained from Cu¹R₅@GN (n=5, S. D.). (c) SERS spectra of Cu²R₁@GN in mitochondrial lysates with the addition of Cu²⁺ with various concentrations (0.2, 0.3, 0.5, 1.0, 2.0, 5.0, 10.0, 15.0, 20.0, 25.0, 28.0 and 30.0 μM). (d) Plots of SERS intensity ratios (I₂₂₃₈/I₂₁₅₅) versus the concentration of Cu²⁺ obtained from Cu²R₁@GN (n=5, S. D.).

Q2-4. The authors investigate copper complexes with Cu¹R₅ and Cu²R₁ through mass spectrometry. However, despite MALDI being a soft technique, mass spectrometry may not accurately represent the species actually present in solution. The formation of the ML complex seems plausible for the Cu¹R₅ ligand, but the 1:1 species formation is not as straightforward in the case of Cu²R₁.

Perhaps UV-vis spectroscopy, employing the method of molar ratio, could be more indicative (clearly at physiological pH or in a mixed solvent if the aqueous solubility of the ligand or the complex/ligand is not optimal).

A2-4. We thank the reviewer for his/her valuable suggestions. The mole ratio method which by using absorbance measurements was further carried out to find the stoichiometry composition of the complex formed. As shown in Figure R2-4a, different amounts of Cu^+ aqueous solution were added to a cuvette containing $20.0 \mu\text{M}$ of Cu^1R_5 ligand dissolved in an acetonitrile (CH_3CN)/PBS buffer (1:1, v/v; 20 mM PBS buffer, $\text{pH} = 7.40$) mixture. A new band around 386 nm was observed after addition of Cu^+ , this absorption band could be attributed to the extension of the electron conjugation system by the chelate structure formed after the coordination of the metal ion with the ligand, which resulted in the emergence of new absorption peaks at longer wavelengths (*Inorg. Chem.* 2011, 50, 1213-1219). In addition, an iso-absorption point was observed at 330 nm , which appeared between the peak maximum of Cu^1R_5 (312 nm) and the peak maximum of the complex (386 nm), suggesting that it was formed by a single equilibrium step (*J. Fluoresc.* 2023, 33, 1003-1015). More importantly, with the concentration of Cu^+ increased from 0 to $20.0 \mu\text{M}$, the intensity of the absorption band around 386 nm was gradually increased. No obvious spectrophotometric change was observed after the concentration of Cu^+ was higher than $20.0 \mu\text{M}$. As summarized in Figure R2-4b, the stoichiometry ratio between Cu^+ ion and Cu^1R_5 ligand was determined as 1:1.

On the other hand, the stoichiometry ratio between Cu^{2+} ion and Cu^2R_1 ligand was also obtained by using a similar method. As shown in Figure R2-4c, a new band around 404 nm was observed upon the addition of Cu^{2+} , and an iso-absorption point was observed at 362 nm , which appeared between the peak maximum for Cu^2R_1 ($\sim 306 \text{ nm}$) and the peak maximum of the complex (410 nm), suggesting that it was formed by a single equilibrium step. The stoichiometry ratio between Cu^{2+} ion and Cu^2R_1 ligand was determined as 1:1 (Figure R2-4d). The discussion was added in the revised manuscript (Page 7, Lines 26-30, and Page 8, Lines 1-15); Figure R2-3 was added in the revised Supporting Information (Page S22, Supplementary Fig. 30).

Figure R2-4. (a) Absorption spectra of Cu^1R_5 ($20\ \mu\text{M}$) upon titration with $[\text{Cu}(\text{CH}_3\text{CN})_4][\text{PF}_6]$ from $0\ \mu\text{M}$ (0 eq.) to $30\ \mu\text{M}$ (1.5 eq.) ($0, 1.5, 2.0, 4.0, 6.0, 8.0, 10.0, 12.0, 16.0, 20.0, 24.0, 25.0$ and $30.0\ \mu\text{M}$) in a $\text{CH}_3\text{CN}/\text{PBS}$ buffer (1:1, v/v; $20\ \text{mM}$ PBS buffer, $\text{pH} = 7.40$) mixture. The peak associated with formation of the complex is at $\lambda_{\text{max}} = 386\ \text{nm}$. (b) A plot of absorbance at $386\ \text{nm}$ as a function of mole ratio ($\text{Cu}^+ : \text{Cu}^1\text{R}_5$) where the tangents intersect at the metal: complex ratio (Error bars: S. D., $n=5$). (c) Absorption spectra of Cu^2R_1 ($20\ \mu\text{M}$) upon titration with $\text{CuCl}_2 \cdot 2\text{H}_2\text{O}$ from $0\ \mu\text{M}$ (0 eq.) to $30\ \mu\text{M}$ (1.5 eq.) ($0, 1.5, 2.0, 4.0, 6.0, 8.0, 10.0, 12.0, 16.0, 20.0, 24.0, 25.0$ and $30.0\ \mu\text{M}$) in a $\text{CH}_3\text{CN}/\text{PBS}$ buffer (1:1, v/v; $20\ \text{mM}$ PBS buffer, $\text{pH} = 7.40$) mixture. The peak associated with formation of the complex is at $\lambda_{\text{max}} = 404\ \text{nm}$. (d) A plot of absorbance at $410\ \text{nm}$ as a function of mole ratio ($\text{Cu}^{2+} : \text{Cu}^2\text{R}_1$) where the tangents intersect at the metal: complex ratio (Error bars: S. D., $n=5$).

Q2-5. Analyzing how the hydrodynamic radius varies during titration with copper(I)/copper(II) could provide insights into various particle aggregation phenomena. These aggregations might occur at the mitochondrial level, subsequently influencing the signal detected for metal quantification.

A2-5. Following the reviewer's suggestions, we supplemented the characterization results of dynamic light scattering (DLS) to estimate

hydrodynamic radius changes of the developed CuPM@GN probe during titration with copper (I) / copper (II). DLS results of the developed CuPM@GN probe were obtained by adding different concentrations of Cu⁺ (1.0, 2.0, 5.0, 10.0, 12.0, 14.0 μM) and Cu²⁺ (1.0, 2.0, 5.0, 10.0, 12.0, 16.0 μM) simultaneously within the linear detection range in aqueous at pH 7.40. As shown in Figure R2-5, the hydrodynamic diameter of individual CuPM@GN probe was estimated as 87.30±1.69 nm. With increasing concentration of copper ions, the hydrodynamic diameters of CuPM@GN were increased by 21.84%-49.43%, which were smaller than those reported in the literatures (> ~115%) (*Langmuir* 2013, 29, 7661-7673; *Environ. Sci.: Nano*, 2016, 3, 567-577; *J. Nanopart. Res.* 2008, 10, 321-332). Importantly, no obvious aggregation in the nanocomposite was observed combined with the size distribution after the dropwise addition of copper ions. These results demonstrated good stability of our developed CuPM@GN probe during copper ions sensing. The corresponding discussion was added in the revised manuscript (Page 7, Lines 18-23); Figure R2-5 was added in the revised Supporting Information (Page S21, Supplementary Fig. 29).

Figure R2-5. Hydrodynamic diameters of CuPM@GN (0.44 mg mL⁻¹) upon addition of both Cu⁺ (a-g: 1.0, 2.0, 5.0, 10.0, 12.0, 14.0 μM) and Cu²⁺ (a-g: 1.0, 2.0, 5.0, 10.0, 12.0, 16.0 μM) in PBS buffer (20 mM, pH=7.40).

Q2-6. In the paragraph titled "Chemicals and Reagents," it is not clear which Cu²⁺ salt was used for titrations to assess the probe's responsiveness to Cu²⁺ concentration. Another important piece of information could pertain to the Cu²⁺ solution: whether it was purchased as is at a known concentration (potentially diluted), or if the exact concentration of the solution was determined after preparation by weighing a salt.

A2-6. $\text{CuCl}_2 \cdot 2\text{H}_2\text{O}$ was used in our work. The standard sample of Cu^{2+} was obtained by quickly and accurately weighing the purchased $\text{CuCl}_2 \cdot 2\text{H}_2\text{O}$ solid drug under nitrogen protection and diluting it with water. The actual weight of the solid was converted into the molar amount, combined with the total volume after dilution to determine the concentration. The corresponding description was added in the revised Supporting Information (Page S3, Lines 26-28).

Q2-7. Supplementary Fig. 1: The authors should review and correct the caption for Figure 1, panel b. "The synthesis procedures of molecules. a, The synthesis procedures of Cu^1R_5 . b, The synthesis procedures of Cu^1R_5 ."

A2-7. The caption of Supplementary Fig. 1b was corrected to "The synthesis procedures of Cu^2R_1 ". (Page S7, Line 9)

Q2-8. Supplementary Fig 3-5-6-8-9-12-16: It is recommended to zoom in or cut the x-axis to clearly show and attribute the peaks.

A2-8. As suggested by the reviewer, all images in Supplementary Fig. 3-5-6-8-9-12-16 have the x-axis enlarged to clearly show the peaks.

Figure R2-6. ^{13}C NMR of Compound 2. ^{13}C NMR (500 MHz, CDCl_3) δ (ppm): 166.30, 138.78, 132.90, 132.42, 130.39, 129.57, 128.71, 126.35, 121.59, 82.40, 80.88, 76.57, 74.28, 52.35 and 45.60.

Figure R2-7. ¹H NMR of Compound 3. ¹H NMR (500 MHz, CDCl₃) δ (ppm): 8.03-8.01 (d, 2H), 7.60-7.59 (d, 2H), 7.52-7.50 (d, 2H), 7.38-7.36 (d, 2H), 3.94 (s, 3H), 3.67 (s, 2H), 2.76-2.65 (m, 16H), 2.59-2.54 (q, 4H), 1.29-1.26 (t, 6H).

Figure R2-8. ¹³C NMR of Compound 3. ¹³C NMR (500 MHz, CDCl₃) δ (ppm): 166.32, 141.26, 132.60, 132.37, 130.25, 129.54, 128.79, 126.52, 120.18, 83.09, 83.05, 80.45, 80.44, 58.42, 53.97, 52.32, 32.42, 31.78, 30.15, 26.08 and 14.82.

Figure R2-9. ^1H NMR of Cu^1R_5 . ^1H NMR (500 MHz, CDCl_3) δ (ppm): 7.79-7.77 (d, 2H), 7.60-7.59 (d, 2H), 7.51-7.50 (d, 2H), 7.38-7.36 (d, 2H), 6.47-6.45 (t, 1H), 4.27-4.26 (dd, 2H), 3.68 (s, 2H), 2.76-2.65 (m, 16H), 2.59-2.54 (q, 4H), 2.31-2.30 (t, 1H), 1.29-1.26 (t, 6H).

Figure R2-10. ^{13}C NMR of Cu^1R_5 . ^{13}C NMR (500 MHz, CDCl_3) δ (ppm): 166.15, 147.78, 133.80, 132.65, 132.61, 128.83, 127.15, 125.49, 120.51, 79.28, 79.25, 77.26, 76.46, 76.42, 72.08, 58.41, 53.97, 32.42, 31.78, 30.10, 29.70, 26.09 and 14.83.

Figure R2-11. ^1H NMR of Compound 4. ^1H NMR (600 MHz, $\text{DMSO-}d_6$) δ (ppm): 8.51-8.50 (d, 2H), 7.80-7.77 (td, 2H), 7.51-7.50 (d, 2H), 7.28-7.26 (m, 2H), 3.81 (s, 4H), 3.36-3.35 (d, 2H), 3.23-3.22 (t, 1H).

Figure R2-12. ^{13}C NMR of Cu^2R_1 . ^{13}C NMR (500 MHz, CDCl_3) δ (ppm): 159.43, 148.94, 136.49, 122.98, 122.04, 83.19, 71.12, 70.39, 68.55, 67.43, 66.10, 60.75, 42.53 and 9.61.

Q2-9. Supplementary Fig 27: The concentration of ES should be indicated in the caption for both panel a and panel b. In panel b, it is also necessary to clarify the copper concentration so that the reader has a clear idea upon initial inspection of the figure.

A2-9. The concentration of elesclomol (ES) (0.001, 0.01, 0.1, 0.5, 1, 5 and 10 μM) was added in the Figure caption in Supplementary Figure S37a (Page S26). In addition, the concentrations of copper and elesclomol-Cu (1:1 ratio) (0.001, 0.01, 0.1, 0.5, 1, 5 and 10 μM) were also added in the figure caption in Supplementary Figure S37b (Page S26).

Q2-10. In the section “ Methods”: Line 736. Check the meaning, What is AA?

A2-10. The full name of AA (ascorbic acid) was added in the revised manuscript. (Page 30, Line 17)

Reply to Reviewer 3

This manuscript developed two Raman reporters which can respond Cu^+ and Cu^{2+} , respectively. The authors used the mitochondrial-specific plasmonic nanoprobes for many scenes, such as, oxidative stress, and drug treatment, and show time-dependent changes. This manuscript is well organized and the data support the conclusion. Before it is acceptable, several issues should be addressed.

Q3-1. In the introduction part, the author stated that "SERS probes assembled based on Au-S bonds are easily susceptible to unpredictable aggregation in cells". To support this, some references should be added.

A3-1. According to the reviewer's suggestions, the related references (*Langmuir* 2012, 28, 4464-4471; *Angew. Chem., Int. Ed.* 2018, 57, 5306-5309; *Langmuir* 2019, 35, 13031-13039) were added in the revised manuscript. (Page 3, Line 5, References 28-30)

Q3-2. Line 301-305, the sentence "Previous literatures has reported that intracellular copper ions were almost bound and stored by various metal-binding proteins rather than free copper ions, to maintain copper homeostasis and avoid free copper ions-involved oxidative stress damage in cells. 38, 39 However, we found a remarkable accumulation of mitochondrial copper ions during ischemia." is wired. The point of the literature is to note that the storage state is metal-binding protein. I didn't get why the author emphasised the accumulation in mitochondria. Is that mean that no metal-binding protein in mitochondria? Please rewrite it here.

A3-2. The distribution of copper ions within the cell was transported and precisely regulated by a series of proteins (*Cell Death Dis.* 2023, 14, 105) due to the high cytotoxicity of copper ion free radicals. Bound copper ions were mainly stored in the mitochondria and endoplasmic (*Acta Pharmacol. Sin.* 2023, 44, 2091-2102). In particular, mitochondrion was an important repository of copper in cells, and mitochondrial physiological functions were also strictly regulated by copper ions (*Cell Rep.* 2015, 10, 933-943). Mitochondrion was the main energy producers in cells and maintained the redox balance within cells. Meanwhile, the increase of free Cu^+ and Cu^{2+} in mitochondrion would cause increased oxidative stress. When neurons were treated with hypoxia and sugar deficiency (OGD) to simulate an ischemic state, the mitochondrion underwent homeostatic imbalance under stress which led to changes in protein structure and function. Therefore, the accumulation of free copper ions in cells mainly came from the release of copper-containing proteins under stress conditions,

which was closely related to mitochondrial function and oxidative stress in mitochondrion.

The sentence was rewritten as “Previous literatures have reported that intracellular copper ions were almost bound and stored by various metal-binding proteins rather than free copper ions, to maintain copper homeostasis and avoid free copper ions-involved oxidative stress damage, especially in mitochondria, due to mitochondria are important copper reservoirs in cells, and physiological functions of mitochondria are strictly regulated by copper ions and oxidative stress (*Cell Rep.* 2015, 10, 933-943). However, we found a remarkable accumulation of mitochondrial copper ions during ischemia” (Page 12, Lines 27-30, and Page 13, Lines 12-16).

Q3-3. The characterization of the SERS probe, some details are missing, e.g., Zeta potential, UV-vis spectra, TEM image, etc.

A3-3. According to the suggestions proposed by the reviewer, zeta potential, UV-vis spectra and TEM image characterizations of the developed SERS probe were further added. Firstly, zeta potentials of GNs, various functionalization intermediates and CuPM@GN probe were measured at physiological pH 7.4. As shown in Figure R3-1a, zeta potential of individual GNs was obtained as -36.1 ± 1.2 mV ($n=5$, S. D.), which can be attributed to the abundance of negative charges provided by citrate salts on the surface of GNs. In addition, zeta potentials of ETPP, Cu¹R₅ or Cu²R₁ functionalized GNs (ETPP@GN, Cu¹R₅@GN, Cu²R₁@GN) were estimated as -33.8 ± 0.9 , -25.3 ± 2.8 and -30.5 ± 1.6 mV, respectively. Interestingly, zeta potential of TPP functionalized GNs (TPP@GN) was estimated as 26.7 ± 2.3 mV ($n=5$, S. D.). The positive potential value of TPP@GN was a result of the modification with positively charged TPP on the surface of GNs. Moreover, zeta potential of the developed CuPM@GN probe was also estimated as 23.1 ± 2.8 mV ($n=5$, S. D.). It should be noted that particle suspensions with an absolute zeta potential greater than 20 mV are generally considered to have anti-aggregation stability, because the charge repulsion is greater than the van der Waals attraction between the particles (*Int. J. Mol. Sci.* 2021, 22, 5072). The results indicated that the developed CuPM@GN probe has good dispersion.

Then, as the UV-vis spectra shown in Figure R3-1b that the maximum absorption wavelength (λ_{\max}) of GNs was slightly red-shifted (~ 3 nm) after CuPM@GN was formed, suggesting that CuPM@GN was dispersed without significant agglomeration. Moreover, TEM images in Figure R3-1c also showed that the CuPM@GN had clear contours and there was no significant increase in size (Figure R3-1d). The corresponding discussion was added in the revised

manuscript (Page 5, Lines 22-23 and Lines 25-28); Figure R3-1 was added in the revised Supporting Information (Page 19, Supplementary Figs. 25c-d and Page 20, Figure S27).

Figure R3-1. (a) Zeta potentials of (I) GNs, (II) TPP@GN, (III) EETP@GN, (IV) Cu¹R₅@GN, (V) Cu²R₁@GN and (VI) CuPM@GN in 20 mM PBS (pH 7.40), respectively. (b) UV-vis spectra of CuPM@GN and GNs, respectively. (c) TEM image of CuPM@GN. (d) Diameter distributions of CuPM@GN.

Q3-4. DFT calculation should be supplemented for Cu¹R and Cu²R, which are also needed for supporting the band assignments of their complex in response to Cu⁺ and Cu²⁺ (Supplementary Table. 1).

A3-4. According to the reviewer's suggestions, we added the DFT calculations of SERS bands assignment of Cu¹R_x and Cu²R_x before and after addition of Cu⁺ and Cu²⁺, as shown in Tables R3-1 to R3-10. Tables R3-1 to R3-10 were added in the revised Supporting Information (Supplementary Tables 1-10)

Table R3-1. SERS bands assignment of Cu¹R₁ before and after addition of Cu⁺.

Cu ¹ R ₁		Cu ¹ R ₁ + Cu ⁺	
Shift (cm ⁻¹)	Assignment	Shift (cm ⁻¹)	Assignment
1080	C-C stretching, C-H	1083	C-C stretching, C-H
1121	C-H twisting, C-N	1124	C-H twisting, C-N
1152	C-H twisting	1155	C-H twisting
1212	C-C stretching, C-H	1213	C-C stretching, C-H
1248	C-C stretching, C-H	1250	C-C stretching, C-H
1270	C-H bending	1272	C-H bending
1346	C-C bending, C-H	1350	C-C bending, C-H
1385	C-H bending	1387	C-H bending
1468	C-H bending	1469	C-H bending
1497	C-H twisting	1499	C-H twisting
2091	HC≡C- stretching	2091	HC≡C- stretching
2236	-C≡C-C≡C- stretching	2238	-C≡C-C≡C- stretching

Table R3-2. SERS bands assignment of Cu¹R₂ before and after addition of Cu⁺.

Cu ¹ R ₂		Cu ¹ R ₂ + Cu ⁺	
Shift (cm ⁻¹)	Assignment	Shift (cm ⁻¹)	Assignment
1037	Ring breathing	1032	Ring breathing
1120	C-N stretching, C-H	1122	C-N stretching, C-H
1131	C-H bending, C-C	1134	C-H bending, C-C
1203	C-H bending	1207	C-H bending
1235	C-C stretching	1239	C-C stretching
1319	C-H bending	1323	C-H bending
1358	C-H bending	1361	C-H bending
1375	C-C stretching	1380	C-C stretching
1468	C-H bending	1470	C-H bending
1496	C-H bending	1499	C-H bending
1550	C-C stretching, C-H	1557	C-C stretching, C-H
1606	Ring stretching	1614	Ring stretching
2105	HC≡C- stretching	2105	HC≡C- stretching
2226	-C≡C-C≡C- stretching	2227	-C≡C-C≡C- stretching

Table R3-3. SERS bands assignment of Cu¹R₃ before and after addition of Cu⁺.

Cu ¹ R ₃		Cu ¹ R ₃ + Cu ⁺	
Shift (cm ⁻¹)	Assignment	Shift (cm ⁻¹)	Assignment
1036	Ring breathing	1029	Ring breathing
1072	C-N stretching, C-C	1077	C-N stretching, C-C
1130	C-N stretching, C-H twisting,	1134	C-N stretching, C-H twisting,
1205	C-H bending	1210	C-H bending
1321	C-H twisting	1326	C-H twisting
1362	C-C stretching, C-H bending	1363	C-C stretching, C-H bending
1374	C-H bending	1379	C-H bending
1417	C-H bending, C-C stretching	1419	C-H bending, C-C stretching
1475	C-N stretching, C-H twisting	1477	C-N stretching, C-H twisting
1496	C-H bending	1501	C-H bending
1550	C-C stretching, C-H bending	1557	C-C stretching, C-H bending
1605	Ring stretching	1612	Ring stretching
1673	C=O stretching	1674	C=O stretching
2114	HC≡C- stretching	2114	HC≡C- stretching
2222	-C≡C-C≡C- stretching	2224	-C≡C-C≡C- stretching

Table R3-4. SERS bands assignment of Cu¹R₄ before and after addition of Cu⁺.

Cu ¹ R ₄		Cu ¹ R ₄ + Cu ⁺	
Shift (cm ⁻¹)	Assignment	Shift (cm ⁻¹)	Assignment
1036	Ring breathing	1029	Ring breathing
1095	C-N stretching, C-C	1098	C-N stretching, C-C
1116	C-N stretching, C-H twisting	1118	C-N stretching, C-H twisting
1133	C-C stretching	1136	C-C stretching
1204	C-H bending	1209	C-H bending
1283	C-H twisting	1287	C-H twisting
1339	C-C stretching, C-H bending	1345	C-C stretching, C-H bending
1354	C-H bending	1359	C-H bending
1462	C-N stretching, C-H twisting	1467	C-N stretching, C-H twisting
1483	C-H bending	1485	C-H bending
1549	C-C stretching, C-H bending	1554	C-C stretching, C-H bending
1596	Ring stretching	1599	Ring stretching
1687	C=O stretching	1688	C=O stretching
2103	HC≡C- stretching	2103	HC≡C- stretching
2220	-C≡C-C≡C- stretching	2221	-C≡C-C≡C- stretching

Table R3-5. SERS bands assignment of Cu¹R₅ before and after addition of Cu⁺.

Cu ¹ R ₅		Cu ¹ R ₅ + Cu ⁺	
Shift (cm ⁻¹)	Assignment	Shift (cm ⁻¹)	Assignment
1038	Ring breathing	1028	Ring breathing
1114	C-N stretching, C-C	1122	C-N stretching, C-C
1201	C-H bending	1208	C-H bending
1337	C-C stretching, C-H bending	1342	C-C stretching, C-H bending
1349	C-H bending	1353	C-H bending
1442	C-N stretching, C-H twisting	1451	C-N stretching, C-H twisting
1537	C-C stretching, C-H bending	1543	C-C stretching, C-H bending
1595	Ring stretching	1597	Ring stretching
1604	Ring stretching	1613	Ring stretching
1691	C=O stretching	1692	C=O stretching
2121	HC≡C- stretching	2121	HC≡C- stretching
2213	-C≡C-C≡C- stretching	2215	-C≡C-C≡C- stretching

Table R3-6. SERS bands assignment of Cu²R₁ before and after addition of Cu²⁺.

Cu ² R ₁		Cu ² R ₁ + Cu ²⁺	
Shift (cm ⁻¹)	Assignment	Shift (cm ⁻¹)	Assignment
1014	Ring breathing	1008	Ring breathing
1074	C-C stretching, C-H bending	1076	C-C stretching, C-H bending
1160	C-N stretching, C-H twisting	1163	C-N stretching, C-H twisting
1215	C-C stretching, C-H bending	1217	C-C stretching, C-H bending
1238	C-C stretching, C-H bending	1239	C-C stretching, C-H bending
1269	C-C stretching	1271	C-C stretching
1310	C-H bending, C-C bending	1313	C-H bending, C-C bending
1385	C-H bending, C-C bending	1385	C-H bending, C-C bending
1448	C-H bending	1449	C-H bending
1498	C-H bending, C=C stretching	1501	C-H bending, C=C stretching
1617	Ring stretching	1624	Ring stretching
2097	HC≡C- stretching	2097	HC≡C- stretching
2238	-C≡C-C≡C- stretching	2239	-C≡C-C≡C- stretching

Table R3-7. SERS bands assignment of Cu^2R_2 before and after addition of Cu^{2+} .

Cu^2R_2		$\text{Cu}^2\text{R}_2 + \text{Cu}^{2+}$	
Shift (cm^{-1})	Assignment	Shift (cm^{-1})	Assignment
1017	Ring breathing	1011	Ring breathing
1038	Ring breathing	1035	Ring breathing
1077	C-C stretching, C-H bending	1078	C-C stretching, C-H bending
1141	C-N stretching, C-H bending	1142	C-N stretching, C-H bending
1167	C-C bending, C-H bending	1169	C-C bending, C-H bending
1198	C-C stretching, C-H twisting	1199	C-C stretching, C-H twisting
1231	C-H twisting	1233	C-H twisting
1323	C-C bending, C-H bending	1325	C-C bending, C-H bending
1375	C-H bending, C-C stretching	1375	C-H bending, C-C stretching
1399	C-H bending, C-C bending	1402	C-H bending, C-C bending
1454	C-H bending	1458	C-H bending
1464	C-H bending	1466	C-H bending
1517	C-H bending, C=C stretching	1519	C-H bending, C=C stretching
1604	Ring stretching	1613	Ring stretching
1614	Ring stretching	1621	Ring stretching
2112	$\text{HC}\equiv\text{C}$ - stretching	2112	$\text{HC}\equiv\text{C}$ - stretching
2229	$-\text{C}\equiv\text{C}-\text{C}\equiv\text{C}-$ stretching	2230	$-\text{C}\equiv\text{C}-\text{C}\equiv\text{C}-$ stretching

Table R3-8. SERS bands assignment of Cu^2R_3 before and after addition of Cu^{2+} .

Cu^2R_3		$\text{Cu}^2\text{R}_3 + \text{Cu}^{2+}$	
Shift (cm^{-1})	Assignment	Shift (cm^{-1})	Assignment
1015	Ring breathing	1007	Ring breathing
1036	Ring breathing	1032	Ring breathing
1071	C-C stretching, C-H bending	1072	C-C stretching, C-H bending
1154	C-N stretching, C-H twisting	1154	C-N stretching, C-H twisting
1199	C-H bending	1202	C-H bending
1321	C-C bending, C-H twisting	1325	C-C bending, C-H twisting
1375	C-H bending, C-C stretching	1376	C-H bending, C-C stretching
1416	C-H bending, C-C bending	1417	C-H bending, C-C bending
1429	C-H bending	1433	C-H bending
1476	C-N stretching, C-H twisting	1477	C-N stretching, C-H twisting
1516	C-H bending, C=C stretching	1518	C-H bending, C=C stretching
1603	Ring stretching	1610	Ring stretching
1615	Ring stretching	1622	Ring stretching
1694	C=O stretching	1694	C=O stretching
2127	$\text{HC}\equiv\text{C}$ - stretching	2127	$\text{HC}\equiv\text{C}$ - stretching
2226	$-\text{C}\equiv\text{C}-\text{C}\equiv\text{C}-$ stretching	2227	$-\text{C}\equiv\text{C}-\text{C}\equiv\text{C}-$ stretching

Table R3-9. SERS bands assignment of Cu²R₄ before and after addition of Cu²⁺.

Cu ² R ₄		Cu ² R ₄ + Cu ²⁺	
Shift (cm ⁻¹)	Assignment	Shift (cm ⁻¹)	Assignment
1016	Ring breathing	1008	Ring breathing
1039	Ring breathing	1036	Ring breathing
1078	C-C stretching, C-H bending	1079	C-C stretching, C-H bending
1151	C-N stretching, C-H twisting	1152	C-N stretching, C-H twisting
1202	C-H bending	1205	C-H bending
1357	C-H bending, C-C stretching	1357	C-H bending, C-C stretching
1402	C-H bending	1405	C-H bending
1411	C-H bending, C-C bending	1412	C-H bending, C-C bending
1434	C-H bending	1436	C-H bending
1464	C-H bending	1465	C-H bending
1481	C-N stretching, C-H bending	1482	C-N stretching, C-H bending
1516	C-H bending, C=C stretching	1519	C-H bending, C=C stretching
1595	Ring stretching	1597	Ring stretching
1614	Ring stretching	1621	Ring stretching
1714	C=O stretching	1715	C=O stretching
2105	HC≡C- stretching	2105	HC≡C- stretching
2224	-C≡C-C≡C- stretching	2226	-C≡C-C≡C- stretching

Table R3-10. SERS bands assignment of Cu²R₅ before and after addition of Cu²⁺.

Cu ² R ₅		Cu ² R ₅ + Cu ²⁺	
Shift (cm ⁻¹)	Assignment	Shift (cm ⁻¹)	Assignment
1017	Ring breathing	1008	Ring breathing
1036	Ring breathing	1032	Ring breathing
1077	C-C stretching, C-H bending	1079	C-C stretching, C-H bending
1171	C-N stretching, C-H bending	1172	C-N stretching, C-H bending
1201	C-H bending	1205	C-H bending
1396	C-H bending, C-C stretching	1397	C-H bending, C-C stretching
1402	C-H bending	1404	C-H bending
1412	C-H bending, C-C bending	1414	C-H bending, C-C bending
1435	C-H bending	1438	C-H bending
1481	C-N stretching, C-H bending	1482	C-N stretching, C-H bending
1523	C-H bending, C=C stretching	1526	C-H bending, C=C stretching
1595	Ring stretching	1596	Ring stretching
1604	Ring stretching	1612	Ring stretching
1614	Ring stretching	1623	Ring stretching
1714	C=O stretching	1714	C=O stretching

2125	HC≡C- stretching	2125	HC≡C- stretching
2215	-C≡C-C≡C- stretching	2216	-C≡C-C≡C- stretching

Q3-5. More experimental details for SERS imaging should be added, such as the accumulation time, objective lens, etc. Did you use Raman mapping mode or imaging mode to obtain the imaging data? If the imaging mode you use, please show the filter information of each channel.

A3-5. The SERS measurements and SERS imaging were performed by a confocal Raman spectrometer (inVia Raman microscope, Renishaw, U.K.). A 785 nm laser with a maximum laser power of 300 mW (100%) was used for all the measurements. Baseline subtraction (cubic spline interpolation) and smoothing (Savitzky-Golay, polynomial order 3, smooth window 9) of Raman Spectra were performed using Renishaw WiRE 5.1 software. For SERS measurements of solution samples, a 10× (N.A. = 0.25) microscope objective with a working distance of 17.6 mm and a spot-focused laser were used. The laser power and acquisition time were 30 mW (10%) and 5 s, respectively. For SERS imaging of live cells, the imaging data were obtained under the Raman mapping mode, which included a 50× (N.A. = 0.5) microscope objective with a working distance of 8.2 mm and a spot-focused laser. The laser power and acquisition time were 3.0 mW (1%) and 1 s. The corresponding experimental method was added in the revised manuscript (Page 31, Lines 26-30 and Page 32, Lines 1-7).

Q3-6. In Figure 3a, it seems CuPM@GN shows strong fluorescence (green color). Did this effect impact SERS measurement?

A3-6. The green fluorescence in Figure 3a in the original manuscript is due to the presence of 6-FAM on the surface of CuPM@GN probe. To study the stability of the developed CuPM@GN probe in mitochondria, 6-FAM labeled DNA strand (5'-HC≡C-(T)₁₅AGCGCAGGCC-6-FAM-3') was further conjugated onto CuPM@GN probe as fluorescent indicator. 6-FAM labeled CuPM@GN probe (6-FAM-CuPM@GN) was only used to estimate the stability of CuPM@GN probe by fluorescence co-localization imaging, and not used for SERS measurements. The detailed experimental method was added in the revised manuscript (Page 32, Lines 9-17).

Q3-7. The bands 2213 and 2238 cm⁻¹ seem overlaid to a certain extent. I suggest using the peak fit function to show their intensities. And Figure 2d-f should be re-plotted according to the fitted data.

A3-7. According to the Reviewer's suggestion, peak fitting was performed on the initial results in Figures 2a-c in our original manuscript. As shown in Figure R3-2a, the Raman peaks of the developed CuPM@GN probe can be fitted to two different peaks around 2213 and 2238 cm^{-1} in the absence of copper ions, which belong to the characteristic peak of Cu^1R_5 and Cu^2R_1 , respectively. In addition, the Raman peaks of the developed CuPM@GN probe can also be fitted to two different peaks around 2213 and 2238 cm^{-1} either in the presence of Cu^+ or Cu^{2+} (Figures R3-2b and R3-2c). Notably, the reference Raman peak can also be fitted to one peak around 2155 cm^{-1} regardless of the presence or absence of copper ions. Therefore, peak areas near 2213, 2238 and 2155 cm^{-1} (A_{2213} , A_{2238} and A_{2155}) were used to represent the signal intensities of the developed CuPM@GN probe for Cu^+ and Cu^{2+} as well as the reference signal intensity, respectively.

Based on the peak fitting results, the standard curves between SERS intensity ratios (A_{2213}/A_{2155} or A_{2238}/A_{2155}) and the concentrations of copper ions (Cu^+ or Cu^{2+}) were obtained. As shown in Figure R3-2d, A_{2213}/A_{2155} displayed good linearity with various concentrations of Cu^+ with a slope of -0.140 ± 0.003 , which was similar to that obtained in Figure 2d in our original manuscript (-0.149 ± 0.003). In addition, A_{2238}/A_{2155} also displayed good linearity with various concentrations of Cu^{2+} with a slope of -0.084 ± 0.003 (Figure R3-2e), which was similar to that obtained in Figure 2e in our original manuscript (-0.094 ± 0.002). Moreover, as shown in Figure R3-2f, the slopes obtained for the simultaneous detection of Cu^+ and Cu^{2+} were also similar to the results in Figure 2f in our original manuscript. Therefore, the quantitative curve obtained using the fitted peak area is consistent with the quantitative curve obtained using the peak intensity. The reason for this is the narrow peak shape of the two peaks of the acetylene stretching vibration (2213 and 2238 cm^{-1}), where a Raman shift difference of 25 cm^{-1} was clearly observed.

More importantly, when performing cellular Raman imaging experiments, it is difficult to analyze the data based on the peak fitting results under the mode limitation. According to previous literature and experience, sub-peak fitting is required when two peaks are highly overlapped, forming a large packet peak and losing characteristic information such as the peak position and the number of peaks. In other cases, the Raman peak intensities reflecting the relative abundance of Raman active vibrational modes can be used directly, which is also due to the properties of Raman fingerprints (*Nat. Commun.* 2018, 9, 607).

Figure R3-2. (a-c) The peak splitting results and parameters under the typical ratio relationship of three different intensities. Line (I) was typical Raman spectrum of CuPM@GN (a) and upon addition of 14 μM Cu⁺ (b) and 16 μM Cu²⁺ (c), respectively. Lines II and III were Raman peaks at 2213 cm⁻¹ and 2238 cm⁻¹ after performing peak fitting, respectively, and Line IV was the baseline set during the peak fitting process. Peak areas near 2213, 2238 and 2155 cm⁻¹ named A₂₂₁₃, A₂₂₃₈ and A₂₁₅₅. All peak types were Gaussian and all R-Square (COD) were greater than 0.98 indicating that the fitting was accurate. (d) Plots of SERS intensity ratios (I₂₂₁₃/I₂₁₅₅) before peak fitting (I) and peak areas ratios (A₂₂₁₃/A₂₁₅₅) after peak fitting (II) versus the concentration of coppers (n=5, S. D.). (e) Plots of SERS intensity ratios (I₂₂₃₈/I₂₁₅₅) before peak fitting (I) and peak areas ratios (A₂₂₃₈/A₂₁₅₅) after peak fitting (II) versus the concentration of coppers (n=5, S. D.). (f) Calibration plots of A₂₂₁₃/A₂₁₅₅ and A₂₂₃₈/A₂₁₅₅ after peak fitting versus concentrations of Cu⁺ and Cu²⁺ (n=5, S. D.).

Reviewers' Comments:

Reviewer #1:

Remarks to the Author:

The authors have adequately addressed my previous comments in the revised version, and I find the quality of the manuscript significantly improved. I don't have any further suggestions and would like to recommend it for publication.

Reviewer #2:

Remarks to the Author:

The authors have reviewed the manuscript and conducted various measurements to address the concerns raised by the referees, and indeed, the manuscript has significantly improved. Upon reading the various responses and modifications, two data points still appear less convincing: the binding constants for the Cu1R5-Cu⁺ complex and Cu2R1 with Cu²⁺. This is particularly noteworthy considering the numerous competitive ligands present within the cell with much more significant constants. Have the authors sought values for similar systems in mixed solvents to compare against the obtained values?

Reviewer #3:

Remarks to the Author:

The authors well addressed the comments. I suggest an acceptance of this manuscript.

Reply to Reviewer 2

The authors have reviewed the manuscript and conducted various measurements to address the concerns raised by the referees, and indeed, the manuscript has significantly improved. Upon reading the various responses and modifications, two data points still appear less convincing: the binding constants for the $\text{Cu}^1\text{R}_5\text{-Cu}^+$ complex and Cu^2R_1 with Cu^{2+} . This is particularly noteworthy considering the numerous competitive ligands present within the cell with much more significant constants. Have the authors sought values for similar systems in mixed solvents to compare against the obtained values?

A2-1. We greatly appreciate the reviewer's high evaluation and valuable suggestions. The binding constants for the $\text{Cu}^1\text{R}_5\text{-Cu}^+$ complex and Cu^2R_1 with Cu^{2+} were estimated to be $\sim 5.65 \times 10^4$ and $\sim 5.61 \times 10^4$, respectively. Within the cell, three main factors may interfere the stability of the $\text{Cu}^1\text{R}_5\text{-Cu}^+$ complex and $\text{Cu}^2\text{R}_1\text{-Cu}^{2+}$ complex, including other metal ions, amino acids and proteins.

The effects of other metal ions (10 mM for K^+ and Na^+ , 1 mM for Ca^{2+} and 50 μM for other metal ions such as Mg^{2+} , Mn^{2+} , Fe^{2+} , Fe^{3+} , Co^{2+} , Ni^{2+} , Zn^{2+} , Pd^{2+} , Cd^{2+} , Pb^{2+}) on the stability of $\text{Cu}^1\text{R}_5\text{-Cu}^+$ complex and $\text{Cu}^2\text{R}_1\text{-Cu}^{2+}$ complex were investigated in the original manuscript (Figure S33). The concentrations of these metal ions were chosen based on physiological concentrations (*ACS Nano 2018, 12, 12357-12368*). Seldom effect ($< 8.2\%$, which was the biggest interference from Zn^{2+}) was observed for other metal ions on the responses of Cu^1R_5 and Cu^2R_1 towards Cu^+ and Cu^{2+} , respectively.

In addition, amino acids in cells may compete with ligands for binding to copper ions due to their functional groups such as amino, carboxyl, or sulfhydryl groups. The binding constants amino acids and copper ions reported in the literature were about $10^2\text{-}10^4$ in physiological pH (pH 6.5-7.8) (*Microchem. J. 1997, 57, 379-388*; *J. Coord. Chem. 2011, 64, 281-292*; *Transition Met. Chem. 1995, 20, 225-227*; *J. Am. Chem. Soc. 1970, 92, 18, 5365-5372*). Therefore, the binding constants of the complex between amino acids and copper ions

were lower than those of $\text{Cu}^1\text{R}_5\text{-Cu}^+$ complex (5.65×10^4) and $\text{Cu}^2\text{R}_1\text{-Cu}^{2+}$ complex (5.61×10^4) due to the limited coordination sites of amino acids. Moreover, as we proved in the original manuscript (Figure S33), addition of the amino acids with physiological concentrations (10 μM for Thr, Gly, Ala, Glu, Leu, Lys, Phe, Tyr and Cys, and 500 μM for GSH) had no obvious impact (< 8.9%) on the responses of Cu^1R_5 and Cu^2R_1 towards Cu^+ and Cu^{2+} , respectively.

In addition, there were copper-containing proteins called copper ion pools in mitochondria, such as metallothionein (MT), copper-zinc superoxide dismutase (Cu, Zn-SOD), cytochrome C oxidase 17 (Cox 17), and synthesis of cytochrome c oxidase 1 (SCO1), which preferentially bound free copper ions to maintain the structure and function of copper-containing proteins under physiological conditions (*J. Inorg. Biochem.* 1990, 39, 137-148; *Science*, 1999, 284, 805). The binding constants of the complexes formed between copper-containing proteins and copper ions were higher than 10^{10} , (*J. Biol. Chem.* 1993, 268, 21533-21537; *J. Biol. Inorg. Chem.* 2002, 7, 327-337), much higher than those of $\text{Cu}^1\text{R}_5\text{-Cu}^+$ complex and $\text{Cu}^2\text{R}_1\text{-Cu}^{2+}$ complex. Importantly, the copper ion binding sites of copper-containing proteins were preoccupied by copper ions, which suggested that copper-containing proteins had negligible effects on copper determination by the developed $\text{Cu}^1\text{R}_5\text{-Cu}^+$ and $\text{Cu}^2\text{R}_1\text{-Cu}^{2+}$ complexes. Moreover, as shown in Fig. R1-1a, the effects of MT (20 $\mu\text{g mL}^{-1}$), Cu, Zn-SOD (20 $\mu\text{g mL}^{-1}$), Cox 17 (5 $\mu\text{g mL}^{-1}$) and SCO1 (5 $\mu\text{g mL}^{-1}$) with physiological concentrations on the stability of $\text{Cu}^1\text{R}_5\text{-Cu}^+$ complex and $\text{Cu}^2\text{R}_1\text{-Cu}^{2+}$ complex were investigated. Almost no effect (< 6.1%) was observed for copper-containing proteins on the responses of Cu^1R_5 and Cu^2R_1 for determination of Cu^+ and Cu^{2+} , respectively.

Furthermore, the effects of common copper-free proteins with physiological concentrations on the stability of $\text{Cu}^1\text{R}_5\text{-Cu}^+$ and $\text{Cu}^2\text{R}_1\text{-Cu}^{2+}$ complexes were also investigated, including albumin from bovine serum (BSA) (1.0 mg mL^{-1}), monoamine oxidase A (MAO-A) (10 $\mu\text{g mL}^{-1}$), monoamine oxidase B (MAO-B) (10 $\mu\text{g mL}^{-1}$), and catechol-O-methyltransferase (COMT)

(5 $\mu\text{g mL}^{-1}$). As shown in Figure R1-1b, seldom effect (< 7.7%) was observed for these copper-free proteins on the responses of Cu^1R_5 and Cu^2R_1 for determination of Cu^+ and Cu^{2+} , respectively. These results proved that the developed Cu^1R_5 and Cu^2R_1 can be used for sensing Cu^+ and Cu^{2+} in mitochondria with high selectivity and accuracy with negligible interference from other metal ions, amino acids, and proteins in the cell. The detailed discussion was added in the revised manuscript (Page 9, Lines 12-13); Figure R1-1 was added in the revised Supporting Information (Page S24, Supplementary Fig. 33).

Figure R1-1. a. Selectivity (Red bars) and competition (Black bars) tests for determination of Cu^+ against proteins. 1: MT (20 $\mu\text{g mL}^{-1}$), 2: (Cu, Zn)-SOD (20 $\mu\text{g mL}^{-1}$), 3: Cox 17 (5 $\mu\text{g mL}^{-1}$), 4: SCO1 (5 $\mu\text{g mL}^{-1}$), 5: BSA (1.0 mg mL^{-1}), 6: MAO-A (10 $\mu\text{g mL}^{-1}$), 7: MAO-B (10 $\mu\text{g mL}^{-1}$), 8: COMT (5 $\mu\text{g mL}^{-1}$). **b.** Selectivity (Blue bars) and competition (Black bars) tests for determination of Cu^{2+} against proteins. 1: MT (20 $\mu\text{g mL}^{-1}$), 2: (Cu, Zn)-SOD (20 $\mu\text{g mL}^{-1}$), 3: Cox 17 (5 $\mu\text{g mL}^{-1}$), 4: SCO1 (5 $\mu\text{g mL}^{-1}$), 5: BSA (1.0 mg mL^{-1}), 6: MAO-A (10 $\mu\text{g mL}^{-1}$), 7: MAO-B (10 $\mu\text{g mL}^{-1}$), 8: COMT (5 $\mu\text{g mL}^{-1}$) (n=5, S. D.).

Reviewers' Comments:

Reviewer #2:

Remarks to the Author:

the manuscript can be accepted in this form.